# Toward economical application of carbon capture and utilization technology with near-zero carbon emission

Kezia Megagita Gerby Langie [1,2,7], Kyungjae Tak[1,7], Changsoo Kim[1], Hee Won Lee[1,3], Kwangho Park[1], Dongjin Kim[1,4], Wonsang Jung[1,3], Chan Woo Lee [2], Hyung-Suk Oh [1,5], Dong Ki Lee [1,6], Jai Hyun Koh [1,3], Byoung Koun Min [1,6], Da Hye Won [1,3] ✉ & Ung Lee [1,3,6] ✉

Carbon capture and utilization technology has been studied for its practical ability to reduce $CO_2$ emissions and enable economical chemical production. The main challenge of this technology is that a large amount of thermal energy must be provided to supply high-purity $CO_2$ and purify the product. Herein, we propose a new concept called reaction swing absorption, which produces synthesis gas (syngas) with net-zero $CO_2$ emission through direct electrochemical $CO_2$ reduction in a newly proposed amine solution, triethylamine. Experimental investigations show high $CO_2$ absorption rates (>84%) of triethylamine from low $CO_2$ concentrated flue gas. In addition, the CO Faradaic efficiency in a triethylamine supplied membrane electrode assembly electrolyzer is approximately 30% (@−200 mA cm$^{-2}$), twice higher than those in conventional alkanolamine solvents. Based on the experimental results and rigorous process modeling, we reveal that reaction swing absorption produces high pressure syngas at a reasonable cost with negligible $CO_2$ emissions. This system provides a fundamental solution for the $CO_2$ crossover and low system stability of electrochemical $CO_2$ reduction.

Carbon capture and utilization (CCU) has been recognized as one of the most promising technologies for mitigating climate change seen in decades because of its capacity for large-scale $CO_2$ reduction. CCU technology is the only group of technologies that can achieve a net-zero emission target by removing direct and balanced emissions[1]. According to the global roadmap, CCU has the potential to reduce carbon emissions by over 7 gigatons by 2030, and the corresponding market size could reach 800 billion USD[2]. However, the industrial application of $CO_2$ conversion technology is still challenging due to its low technological maturity, expensive production cost, and high energy consumption[3]. One way of early settlement of CCU technology

is the sequential conversion of $CO_2$ via synthesis gas (syngas) generation. Sequential production via syngas has the great advantage of producing various chemicals by adjusting the $H_2$-to-CO ratio and employing subsequent reaction processes[4]. However, conventional $CO_2$-derived syngas production methods are highly endothermic, which hinders commercialization and consequently leads to question about their ability to reduce $CO_2$.

Electrochemical $CO_2$ reduction (e$CO_2$R) has attracted much attention as an alternative route for sustainable production of CO or syngas[5–8]. e$CO_2$R provides a pathway for achieving net-zero emission during syngas production by utilizing renewable energy. Nevertheless,

[1]Clean Energy Research Center, Korea Institute of Science and Technology, Seoul 02792, Republic of Korea. [2]Department of Chemistry, Kookmin University, Seoul 02707, Republic of Korea. [3]Division of Energy and Environmental Technology, KIST School, Korea University of Science and Technology (UST), Seoul 02792, Republic of Korea. [4]Department of Materials Science and Engineering, Korea University, Seoul 02841, Republic of Korea. [5]KIST-SKKU Carbon-Neutral Research Center, Sungkyunkwan University, Suwon 16419, Republic of Korea. [6]KU-KIST Graduate School of Energy and Environment, Korea University, Seoul 02841, Republic of Korea. [7]These authors contributed equally: Kezia Megagita Gerby Langie, Kyungjae Tak. ✉e-mail: dahye0803@kist.re.kr; ulee@kist.re.kr

eCO$_2$R processes are not economically attractive because CO$_2$ conversion through eCO$_2$R generally requires expensive CO$_2$ capture and product conditioning processes for purified raw material and product supply. Recently, novel efforts have been reported to overcome the cost-inefficiency of eCO$_2$R technology by developing new systems excluding pre-/post-conditioning processes, such as utilization of low-concentration CO$_2$ (e.g., flue gas)[9–11] and direct conversion of CO$_2$ captured in an amine solution[12,13]. In particular, direct eCO$_2$R in an amine solution is a promising strategy because it could eliminate energy-intensive thermal amine regeneration and consume negligible energy for pressurization. Theoretically, the net-zero CO$_2$ conversion process can be built based on direct conversion of CO$_2$ captured in an amine when the necessary electricity is supplied from a renewable source.

Several attempts have been made to perform eCO$_2$R in commercial CO$_2$-capturing absorbents (e.g., monoethanolamine, diethanolamine, 2-amino-2-methyl-1-propanol, and their mixtures), and their limitations have been discovered[12,13]. These primary and secondary alkanolamine solvents capture CO$_2$ as a carbamate, which inherently contains a strong C-N bond that makes CO$_2$ conversion difficult[14]. Chen et al. made an early attempt to demonstrate the possibility of eCO$_2$R in a 30 wt.% monoethanolamine solution[12]. This process exhibited a moderate CO Faradaic efficiency (FE) of 38.2%, but the authors also found that the active carbon source for the eCO$_2$R was free CO$_2$ in the solution, not captured CO$_2$ as carbamate. Lee et al. conducted eCO$_2$R with captured CO$_2$ in a monoethanolamine solution and first observed that the steric properties of a cation in the Helmholtz layer caused the limited performance of eCO$_2$R[13]. Addition of electrolytes involving alkali metal cations could constitute a method for improving the eCO$_2$R performance. However, this method may be economically unfavorable due to the expensive supply of electrolytes. Li et al. shed light on the direct-capture CO$_2$ conversion system by suggesting bicarbonate as a valid option for converting captured CO$_2$; this process showed a 37% FE for CO at −100 mA cm$^{-2}$ without the addition of supporting electrolyte[15], and the performance has further increased to 95% FE for CO at −100 mA cm$^{-2}$ under 4 atm pressure condition, in a recent follow-up study[16]. In this system, KOH is considered to be a currently available CO$_2$ absorbent. Although KOH is a promising solvent for CO$_2$ capture and conversion, the salt formation and corrosion issues caused by the extremely alkaline condition are considered as major challenges for the commercialization[17–20]. Therefore, a specific method for sustainably supplying bicarbonates from flue gas is required. Most importantly, process design and systematic analyses such as a techno-economic analysis (TEA) or life cycle assessment (LCA) have not been applied in spite of the technological potential of achieving net-zero CO$_2$ conversion and producing high-pressure syngas.

Herein, we present an economically feasible and environmentally benign methodology for CO$_2$ reduction called a reaction swing absorption (RSA). RSA is a new syngas production method that directly utilizes CO$_2$ in flue gas without CO$_2$ capture or product separation processes. Throughout this work, we demonstrate experimentally the capability of CO$_2$ absorption and direct eCO$_2$R of RSA. A conceptual design for RSA is also provided, which shows that RSA can be operated without thermal energy consumption; thus, net-zero CO$_2$ emission is possible depending on the energy supply. TEA and LCA are also provided to highlight future research guidelines required for sustainable chemical production via RSA.

## Results
### Reaction swing absorption
The term RSA refers to processes that selectively separate a target chemical from a gas stream using an absorbent and chemically convert it to desired products. The products are released from the absorbent as the reaction takes place due to solubility differences between reactants and products. For example, the CO$_2$ RSA separates CO$_2$ from flue gas, and subsequent eCO$_2$R in an amine solvent releases high-purity syngas because the product has low solubility in the amine solvent. Figure 1 shows the generalized concept for RSA production of high-pressure syngas (i.e., a CO and H$_2$ gas mixture) from CO$_2$. RSA consists of chemisorption, pressurization, and electrochemical conversion (e-chemical)[21] processes. The chemisorption process generates bicarbonate as an absorbent and captures CO$_2$ from the flue gas. The bicarbonate solution is pressurized with a pump and then selectively reduced to CO in a zero-gap membrane electrode assembly (MEA) electrolyzer. The unreacted CO$_2$ remains in the solution, so a gas product separation process is no longer required.

Figure 1 also describes two conventional syngas production pathways; the reverse water gas shift reaction (RWGS) and gas phase eCO$_2$R[9,13,22]. Both pathways exhibit far more complicated process configurations than CO$_2$ RSA because they require CO$_2$ separation, pressurization, and product separation. In conventional processes, thermal solvent regeneration in CO$_2$ separation and high-temperature syngas generation (i.e., RWGS) consume considerable amounts of thermal energy, so CO$_2$ reduction in these systems shows limited capability[23]. Additionally, considering the low one-path conversion to CO in eCO$_2$R (<50%), an additional process for separating unreacted CO$_2$ and recycling it back to the electrolyzer should be considered to maximize carbon utilization[24]. Furthermore, even the most advanced eCO$_2$R electrolyzers (e.g., a zero-gap MEA electrolyzer) have experienced systematic problems such as carbonate crossover, which causes system instability and low energy efficiency. The RSA successfully overcomes the abovementioned drawbacks while producing high-pressure and purified syngas. We first describe experimental observations for unit operability of the RSA process and present comprehensive comparisons among CO$_2$ RSA, RWGS, and gas phase eCO$_2$R via TEA analysis and LCA.

### Amine selection for selective CO$_2$ absorption as bicarbonate
Alkanolamines are commonly used in industry due to their fast CO$_2$ absorption rates and high capacities, but most of them capture CO$_2$ as carbamate[25,26]. According to previous studies conducted on eCO$_2$R with CO$_2$ captured in amine solutions, CO$_2$ is not easily released from carbamates due to the strong C-N bonding[13,14]. Thus, it is important to find a new amine that captures CO$_2$ in a mild form such as bicarbonate, and directly converts amine-captured CO$_2$. After screening various amines, including primary, secondary, and tertiary amines, we found that triethylamine (TREA) is an ideal solvent for both CO$_2$ capture and bicarbonate utilization, which contains aliphatic groups but not hydroxyl groups. We confirmed the presence of bicarbonate as the major form of CO$_2$ captured in a TREA aqueous solution by nuclear magnetic resonance (NMR) analysis (Fig. S1). In the $^{13}$C-NMR spectrum of CO$_2$ captured in a 3 M TREA/H$_2$O solution, a single peak derived from bicarbonate (and/or carbonate) ion was observed. Carbamate species were not observed. Considering that the CO$_2$ saturated 3 M TREA/H$_2$O is a neutral solution (pH 7–8), it can be deduced that bicarbonate was selectively generated during the absorption of CO$_2$ by TREA according to the chemical equilibrium determined by pH[27]. Figure 2a suggests a simplified mechanism for CO$_2$ capture in a 3 M TREA solution. The mechanism shows an equimolar reaction between CO$_2$ and TREA as bicarbonate generation yields TREA protonation. Note that the theoretical absorption capacity of the TREA is twice that of monoethanolamine, although monoethanolamine has higher reactivity.

The CO$_2$ absorption capacity of TREA solvent was measured with a bench scale absorption column equipped with structure packing (Figs. 2b and S2). We measured the CO$_2$ absorption rates and capacities at liquid feed/gas CO$_2$ feed (L/CO$_2$) ratios ranging from 0.07 to 0.21 with synthesized flue gas containing 3–5% CO$_2$. Figure 2c shows that the higher the L/CO$_2$ ratio at fixed gas flow and CO$_2$ concentration, the

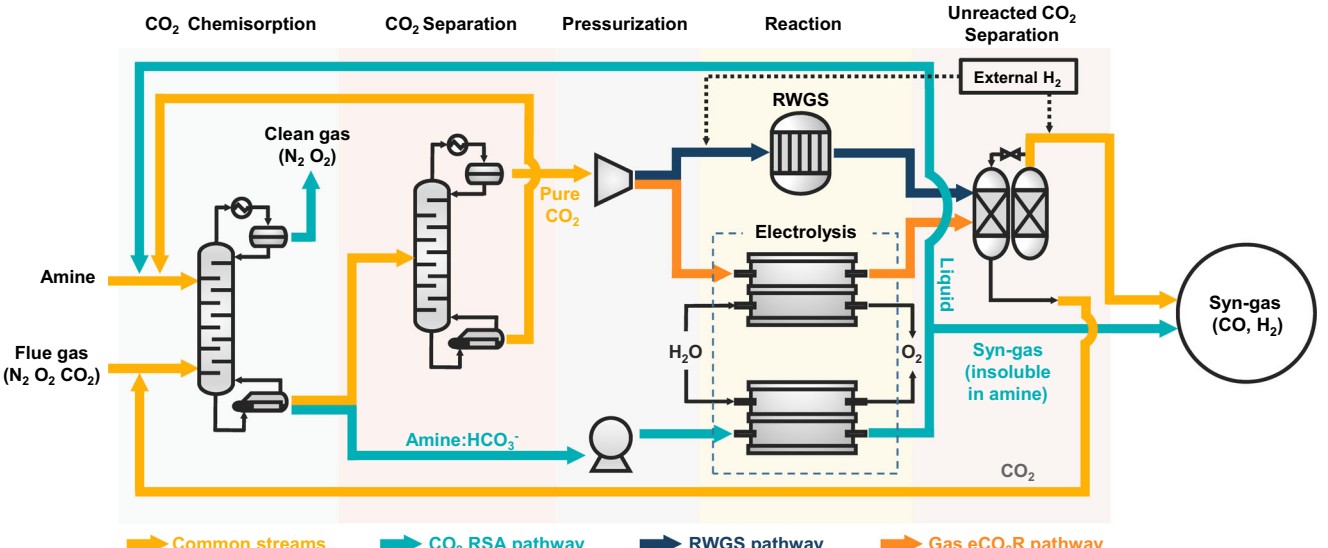

**Fig. 1 | Schematic diagram comparing the RSA pathway with other CCU pathways.** Graphical representation of the reaction swing absorption process. The light blue stream represents the RSA pathway. Compared to the RWGS (dark blue) and gas eCO$_2$R (orange) pathways, the RSA pathway exhibits a simple process configuration, retains the highest portion of electricity, and is potentially capable of providing a viable CCU solution.

greater the CO$_2$ absorption rate but the less CO$_2$ absorption capacity. Figure 2d presents a linear relationship between L/CO$_2$ ratio and CO$_2$ absorption capacity, regardless of liquid flow (2–5 L h$^{-1}$), gas flow (0.5–0.8 m$^3$ h$^{-1}$), and CO$_2$ concentration (3–5%). All CO$_2$ absorption experiments with TREA in this study exhibit high performance, in terms of CO$_2$ absorption rate from 84.5% at fast flue gas flow rate (0.8 m$^3$ h$^{-1}$ with 3% CO$_2$) to 95.1% at slow flow (0.5 m$^3$ h$^{-1}$ with 3% CO$_2$) shown in Fig. S2.

### Desirable system configuration for direct conversion of amine-captured CO$_2$

CO$_2$-captured TREA solution was then supplied to the electrochemical system for CO$_2$-derived syngas generation. The eCO$_2$R system was designed based on the MEA electrolyzer, which is the most suitable system applied commercially (Fig. 3a)[28,29]. The eCO$_2$R system configuration is similar to that of the catholyte-free MEA electrolyzer except that an amine-captured aqueous CO$_2$ solution is provided to the cathode instead of humidified CO$_2$ gas. Since the targeted product is syngas, Ag nanoparticles (a representative catalyst for CO production)[9,30–34] on a carbon supporter (Ag/C) were initially selected as a standard catalyst in developing the system. Ni foam was used as the anode while 1 M KOH was supplied as an anolyte. Between the cathode and anode, a bipolar membrane was positioned to separate the two electrodes and supply proton and hydroxide ions to each electrode (Fig. 3b). Gas products from CO$_2$ conversion were delivered back to a liquid reservoir with the CO$_2$ capture solutions, and then analyzed by gas chromatography (GC). Only gaseous products were detected in this system.

To demonstrate the excellent capability of TREA, eCO$_2$R with various CO$_2$ absorbents such as monoethanolamine and diethanolamine were compared to that with TREA in the abovementioned system configuration (Figs. 3c and S3a). CO$_2$ conversion was evaluated with CO FE at applied constant current densities ranging from −20 to −200 mA cm$^{-2}$ by using chronopotentiometry. CO FEs at all conditions were below 15% when 30 wt.% monoethanolamine and 30 wt.% diethanolamine were used. However, the 3 M TREA outperformed the other amines, demonstrating that it is the best option for a feasible system. Considering that monoethanolamine and diethanolamine transform CO$_2$ into carbamates, bicarbonate formation with TREA is the most favorable for direct conversion of CO$_2$. Moreover,

experiments in which the bicarbonate concentration in TREA was decreased over 6 h of eCO$_2$R confirmed that bicarbonate served as the carbon source for CO production (Supplementary Note 3 and Table S1). Initially, there 2.78 M bicarbonate was present in the TREA solution prior to the reaction, and the concentration linearly decreased to 2.47 M as the reaction progressed in 6 h. This result indicated that bicarbonate was utilized as the major CO$_2$ source. During the eCO$_2$R, CO$_2$ was not detected by GC, implying that unreacted CO$_2$ was efficiently captured as bicarbonate in the TREA solution (Fig. S4).

Syngas production by the RSA system was further improved by cathode development (Figs. 3d and S3b). In a typical catholyte-free MEA system, CO$_2$ is supplied in the gas phase, and thus, hydrophobicity is required for the catalyst electrode to form a three-phase interface, which acts as an active site[35,36]. In contrast, this direct amine-captured CO$_2$ conversion system utilizes CO$_2$ from the liquid electrolyte, so a hydrophilic catalytic electrode would be preferred for sufficient interaction of the catalyst layer with CO$_2$. To simultaneously achieve a suitable electronic structure enabling high CO selectivity for the Ag catalyst as well as hydrophilicity for the carbon supporter, we electrochemically modified the Ag/C into a coral like-structure (coral-Ag/C) through electrochemical oxidation and reduction processes (Fig. 3e)[32]. To identify the effects of the coral-Ag structure and carbon hydrophilicity, additional catalysts, such as (i) Ag nanoparticles (Ag NP) and (ii) coral-structured Ag nanoparticles (coral-Ag), were prepared as a control group, and their characteristics were correlated with their CO production performance (Figs. S5–S10). As expected, the coral-Ag exhibited a higher CO FE than the Ag NPs, and this result clarified that the coral structure was favorable for CO production in the amine-captured CO$_2$ conversion system. Randomly connected 3D coral structure with partially oxidized Ag (Ag$^+$) near the surface provided suitable surface binding affinities for reaction intermediates[32]. More importantly, the coral-Ag/C catalyst achieved an outstanding CO FE of 70% at a current density of −20 mA cm$^{-2}$ and 30% at a high current density of −200 mA cm$^{-2}$ (Fig. 3d). Considering the moderate performance improvement from Ag NP to Ag coral and from Ag NP to Ag/C, the improved performance of coral-Ag/C implies that there may be other factors to contribute to the performance in addition to the structural modification of Ag and the introduction of carbon support. Thus, we suggest that the hydrophilicity of the electrode is a novel

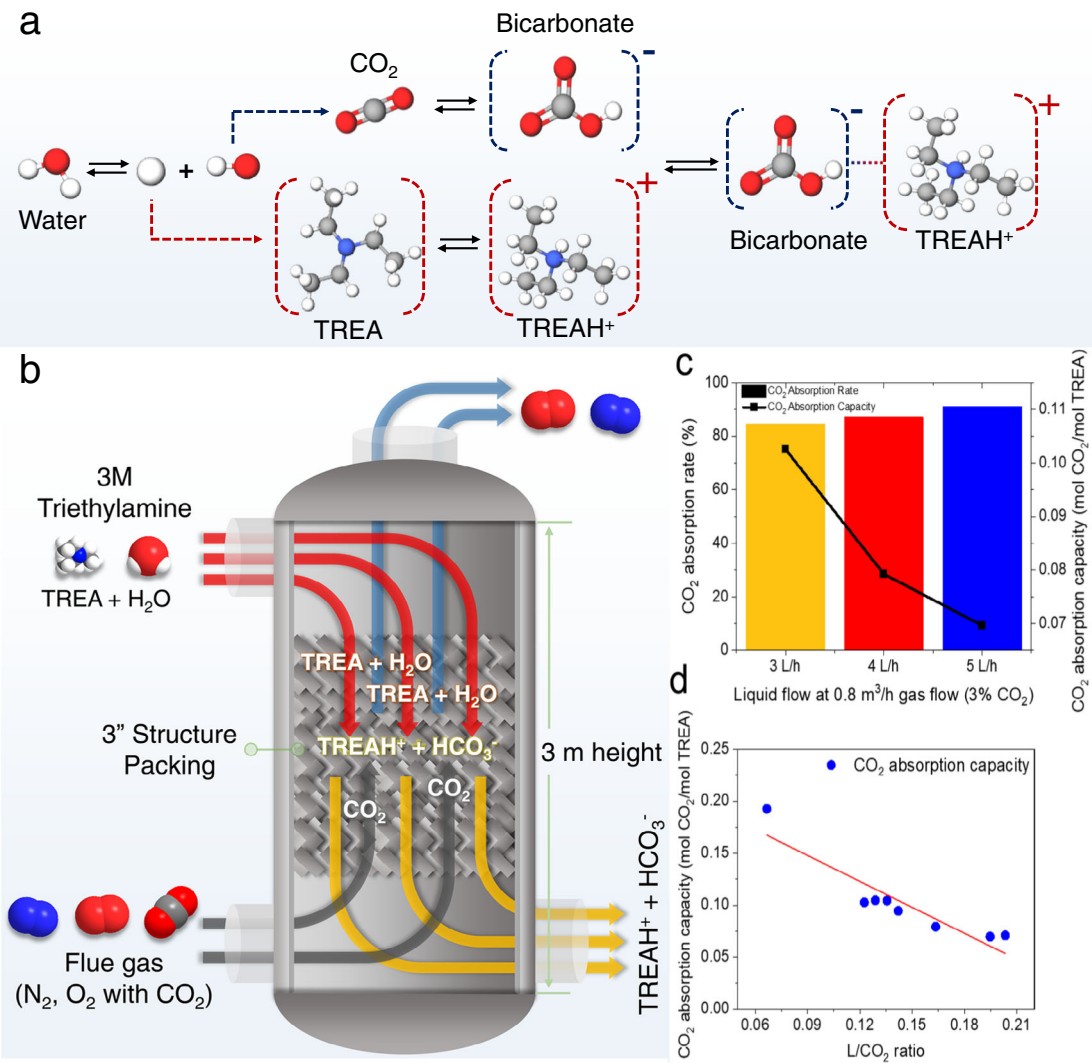

**Fig. 2 | The reaction mechanism for $CO_2$ captured in TREA solution and $CO_2$ absorption performance in 3 M TREA solvent. a** A representation of the proposed $CO_2$ capture process is shown, where the TREA solution captures $CO_2$ by stabilizing the bicarbonate on an equimolar basis. **b** A schematic figure of the $CO_2$ absorption column used for the capture experiments. The column is 3 meters in height with a diameter of 3 inches. **c** $CO_2$ absorption experiment results with varying TREA flow rates at fixed gas flow rate (0.8 $m^3$ $h^{-1}$) and $CO_2$ concentration in gas flow (3% $CO_2$). **d** Effect of $L/CO_2$ ratios on $CO_2$ absorption capacity at various values of TREA flow rates (2–5 L $h^{-1}$), gas flow rates (0.5–0.8 $m^3$ $h^{-1}$), and $CO_2$ concentrations (3–5%). $CO_2$ absorption capacity shows a linear correlation with $L/CO_2$ ratio, regardless of TREA flow rate, gas flow rate, and $CO_2$ concentration.

critical factor determining catalytic performance in the RSA system. As shown from contact angle measurements, coral-Ag/C showed the lowest contact angle among the electrodes (Fig. 3e). Similar contact angles for coral-Ag and Ag NPs implied that the electrochemical oxidation and reduction processes were not influenced by the hydrophilicity of Ag but by that of carbon support, as we expected. This influence might derive from increases in the number of oxygen functional groups present on the carbon supporter during electrochemical processes, as observed via X-ray photoelectron spectroscopy (XPS) analysis (Fig. S11). We further compared the $eCO_2R$ performance according to hydrophobicity of electrodes (Fig. S12). As expected, hydrophobic electrodes showed worse performance with the same Ag catalysts.

System sustainability was secured through selection of an appropriate membrane and confirmed by long-term stability. In the RSA system, since 3 M TREA and 1 M KOH solutions are respectively utilized as catholyte and anolyte, both electrolyte separation for recycling and proton supply for releasing $CO_2$ from bicarbonate are important. In this regard, a bipolar membrane is the best option for

supplying protons to the cathode and hydroxides to the anode, respectively, while guaranteeing negligible ion transfer between the catholyte and anolyte. In cases with other membranes, such as anion exchange and cation exchange membranes, ion transfer inevitably occurs due to the intrinsic ion-conducting properties, and this ion transfer interferes with the long-term use of the electrolytes. In practice, only bipolar membrane systems have exhibited acceptable CO production performance, while other types of membranes have shown poor CO production performance (Figs. 3f and S3c). It was also confirmed that the bipolar membrane prevented bicarbonate crossover to the anode, since no $CO_2$ gas evolution was detected from the anode (Fig. S4). This optimized system configuration (coral-Ag/C cathode and bipolar membrane) demonstrated stable performance with 35% of CO FE, during a 70 h operation of chronopotentiometry experiment at −100 mA $cm^{-2}$ (Fig. 3g).

**Techno-economic analysis and life cycle assessment of RSA**

We carried out a comprehensive TEA and an LCA of RSA using a process model based on the abovementioned experimental results. The

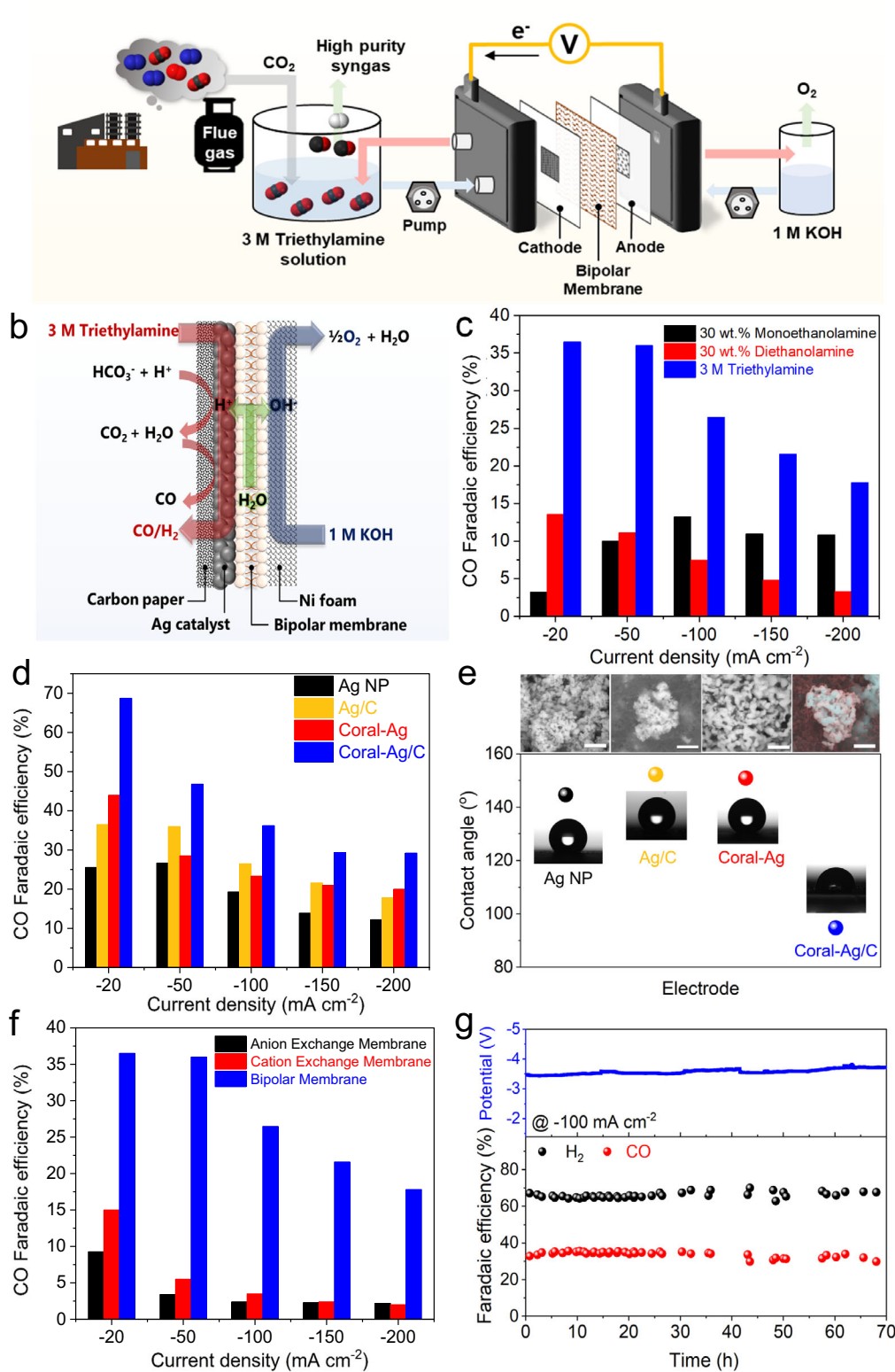

**Fig. 3 | Syngas production with the eCO₂R system. a** Schematic of the eCO₂R system configuration. **b** Schematic of chemical reactions with a cathode, anode, and bipolar membrane. **c** CO FEs for Ag/C measured with various applied current densities in monoethanolamine, diethanolamine, and TREA. **d** CO FEs for Ag NP, Ag/C, coral-Ag, and coral-Ag/C in 3 M TREA. **e** SEM images (scale: 500 nm) and contact angles for all Ag electrodes. **f** CO FEs for Ag/C measured with various membranes, including an anion exchange membrane, cation exchange membrane and bipolar membrane, in 3 M TREA. **g** Long-term CO production represented as the H₂:CO ratio, CO FEs, and cell voltages (blue) for coral-Ag/C at −100 mA cm⁻² in 3 M TREA.

result was compared with those of conventional processes using the RWGS and gas phase $eCO_2R$ (Fig. 1) to confirm the superiority of the proposed RSA process. A global sensitivity analysis (GSA) was also performed to assign cost contributions and global warming potentials (GWPs) and eventually to highlight the crucial factors for further improvement of RSA[37]. Current and optimistic scenarios were compared through analysis of several related factors: (i) the electricity generation cost based on different energy sources, (ii) improvements to the bicarbonate electrolyzer, (iii) purchase cost of $H_2$ produced from an on-site water electrolysis system as an alternative option to satisfy the $H_2$-to-CO ratio of 2 for syngas, and (iv) $CO_2$ capture rate of the absorber. $H_2$ in syngas comes from purchase, from the $CO_2$ electrolyzer as a by-product, or from both. There is competition of $H_2$ supply between the bicarbonate electrolyzer and on-site purchase. Therefore, the third factor (purchase cost of $H_2$) is added for the sensitivity analysis. Whereas the operating conditions of RWGS and gas $eCO_2R$ processes are well known, those of the RSA should be optimized due to its early development stage. As a result, the optimal cell voltage, which is pertinent to the CO FE and current density, was found to minimize the break-even price of syngas. The details of the process modeling methodology and TEA and LCA results are explained in Supplementary Note 4.

As illustrated in Figs. 4a, b, S18 and S19, the RSA process outperforms the other two CCU processes, in terms of operating expenditure (OPEX) and the break-even price of syngas in the optimistic scenario, regardless of energy sources. In the current scenario, all three processes have a similar OPEX and break-even price, although the RSA shows an enormous capital expenditure (CAPEX) due to the inefficient bicarbonate electrolyzer. However, current density can be increased at lowered voltage with higher CO FE, as the performance of bicarbonate electrolysis is improved in the optimistic scenario. This improvement considerably reduces CAPEX of the RSA and converts the $H_2$ supply from the bicarbonate to water electrolyzer. Considering lower cell voltage in water electrolysis than that in bicarbonate electrolysis, the $H_2$ supply change results in the RSA beating other two processes for CAPEX, OPEX, and break-even price in the optimistic scenario. When renewable energy is used, the optimistic break-even prices for syngas will be dropped to $0.65/kg of syngas for wind and $0.56/kg of syngas for solar. These prices can compete with fossil fuel-based syngas processes (Fig. 4f). Although the wind case has a slightly higher syngas price than that in the solar case, wind is a more promising energy source from an environmental perspective (only one-third of the GWP100 value in comparison to the solar case).

Figure 4c presents the system boundaries for LCAs containing $CO_2$ sources within the boundaries. Two important impacts for an LCA of CCU technology are GWP100 and fossil resource scarcity (FRS). In the case of GWP100, the RSA process has a lower impact than the other processes in all cases except the current energy mix case (Fig. S20). The GWP100 of the RSA can be minimized to 0.27 kg $CO_2$ eq./kg syngas (Fig. 4d) when wind energy is used for electricity generation. However, GWP100 is highly sensitive to the energy source, so this value can be increased up to 5.52 kg $CO_2$ eq./kg syngas in the energy mix case (Fig. 4e). The FRS result shows a similar trend to GWP100, because the RSA is highly energy-intensive process.

A GSA was also conducted to support decision-making for establishing appropriate CCU strategies and policies and to provide the priority of research targets. Unlike local sensitivity analyses, such as a one-factor-at-a-time method, GSA changes all uncertain input variables simultaneously and monitors the variances of dependent variables[38]. Thus, a GSA generates Sobol indices, which indicate the impacts of uncertain (input) variables on dependent (output) variables. There are six uncertain variables in the RSA for the solar and wind cases: $CO_2$ capture rate, CO FE improvement, current density improvement, unit cost of the electrolyzer, electricity generation improvement, and additional cost (except electricity cost) when $H_2$ in

syngas is supplied by an on-site water electrolysis system, instead of $H_2$ production in the bicarbonate electrolyzer as by-product. For the energy mix case, portions of solar and wind energies in electricity generation mix are added to the uncertain variable set.

Figure 4g compares the 1st order Sobol indices of the break-even price of syngas and the GWP100 for the RSA process under a given range of uncertain variables from current to optimistic scenarios. When wind energy is used for the RSA, the most sensitive economic factor is the electricity generation price (electricity generation improvement in Fig. 4g). Wind energy is well-known for its very low $CO_2$ emission, so electricity becomes no longer a major environmental issue. Consequently, uncaptured $CO_2$ to be released to air from the absorber is the main contributor to $CO_2$ emission when wind energy is used. This result indicates that the near net-zero $CO_2$ emission is achievable as the $CO_2$ capture rate increases in the chemisorption process. For the energy mix case, no factor dominantly impacts the break-even price. From an environmental perspective, the additional cost except electricity cost to the on-site water electrolysis system is the most significant factor. In other words, the improvement of water electrolysis infrastructure and technology economically attracts this system for $H_2$ production and substantially reduces electricity usage. Consequently, the amount of emitted $CO_2$ in the energy mix case can be largely decreased by this improvement. The detail TEA, LCA, and GSA results for different scenarios and cases can be seen in Supplementary Note 4.

## Discussion

We report RSA as a potential net-zero $CO_2$ emission methodology for syngas production from $CO_2$. By utilizing a bicarbonate-generating amine solvent, TREA, RSA successfully captures $CO_2$ from flue gas and selectively reduces it to syngas via direct electrochemical $CO_2$ reduction. The $CO_2$ absorption capability of TREA was measured with a bench scale absorber and a scalable system configuration for $eCO_2R$, and this further supported the feasibility of RSA in industrial applications. After optimization of variables in the $eCO_2R$ system, such as catalyst, electrolyte, and membrane, RSA produced high-purity syngas composed of 30 to 70% CO with a balance of $H_2$, depending on applied current densities ranging from $-200$ mA cm$^{-2}$ to $-20$ mA cm$^{-2}$. It is worth noting that the impact of trace amounts of impurities, such as $NO_3^-$, on the bicarbonate electrolysis system should be investigated before upscaling the process[16]. A TEA based on the experimental results showed that RSA is the most promising CCU process with the lowest $CO_2$ emissions, and it outperforms current CCU processes.

CCU technology has been studied substantially but commercialization is still limited by many challenges and difficulties originating from the high stability of $CO_2$. If syngas, a product of the endothermic reactions of $CO_2$, can be produced economically, then it is possible that CCU technology can be industrialized because the downstream processes are exothermic (Fig. 5). For example, methanol, ethanol, and dimethyl ether synthesis and the Fischer-Tropsch reaction are highly exothermic, implying that these downstream processes can be operated without excessive energy consumption. Considering the technological maturity of downstream processes utilizing syngas, we believe that RSA can change the paradigm for chemical processes by realizing the eco-friendly and cost-effective production of syngas from $CO_2$. In Fig. 5, energy-intensive syngas production accounts for the highest proportion in terms of cost and energy consumption. Therefore, employing an environmentally benign and economically feasible process of syngas production could accelerate the commercialization of CCU technology. Such acceleration may be achieved through the RSA process proposed in this study because it requires neither an energy-intensive step for amine solvent regeneration nor a separation system of unreacted $CO_2$. According to the modeling results of this study, it is crucial to improve the bicarbonate electrolyzer used in RSA and integrate the system with renewable energy sources in order to

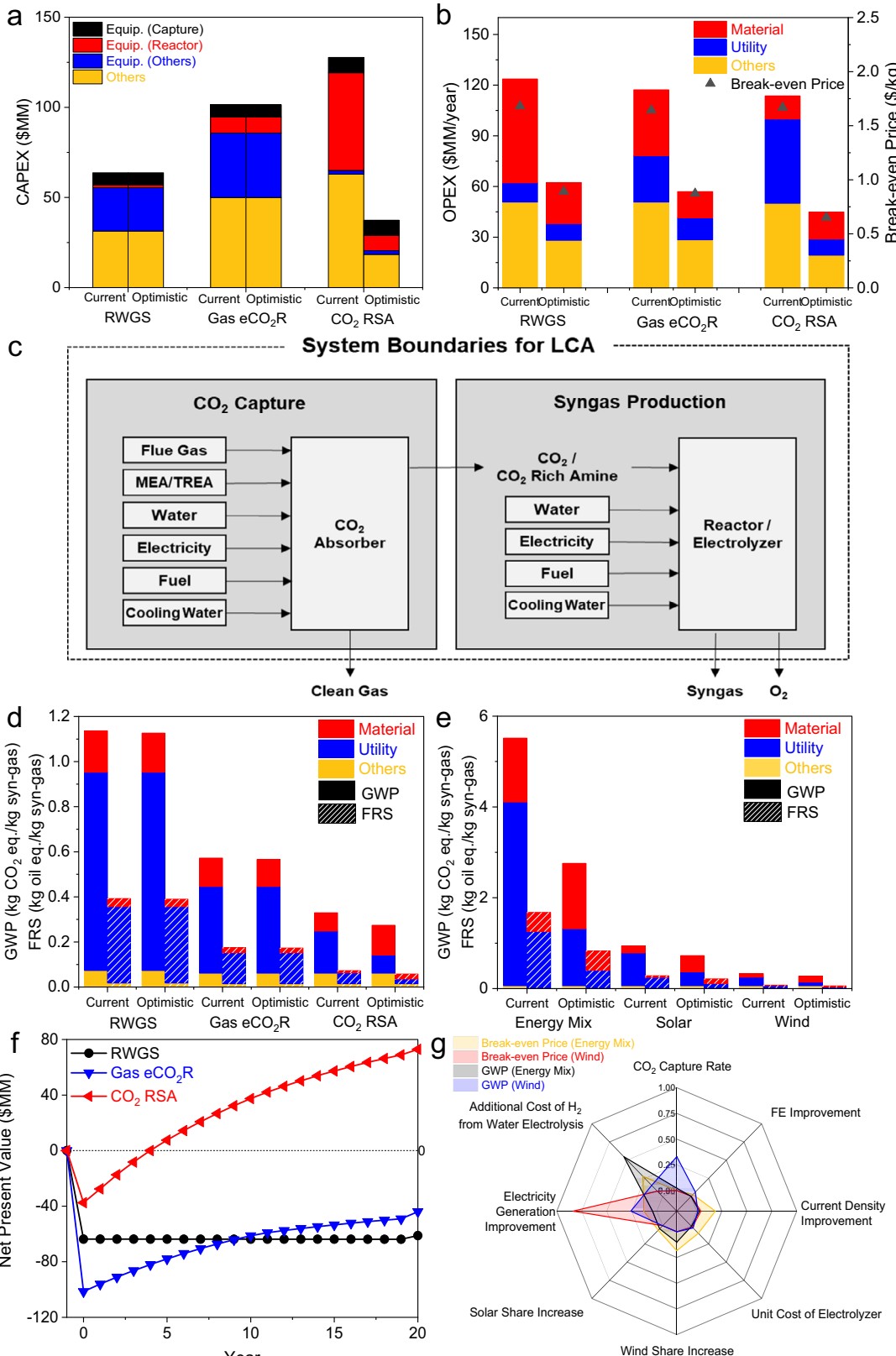

**Fig. 4 | Results of TEA, LCA, and GSA for the wind case. a** Capital expenditures for three processes. **b** Operating expenditures and break-even prices for syngas from the three processes. **c** Graphical representation of LCA system boundaries. **d** LCA results for the three processes. The left and right bars in each scenario represent global warming potential (GWP) and fossil resource scarcity (FRS), respectively. **e** LCA results for the RSA process with different cases and scenarios. **f** Cash flow charts for an optimistic scenario with a selling price of $0.8/kg syngas. **g** 1st order Sobol indices from global sensitivity analysis.

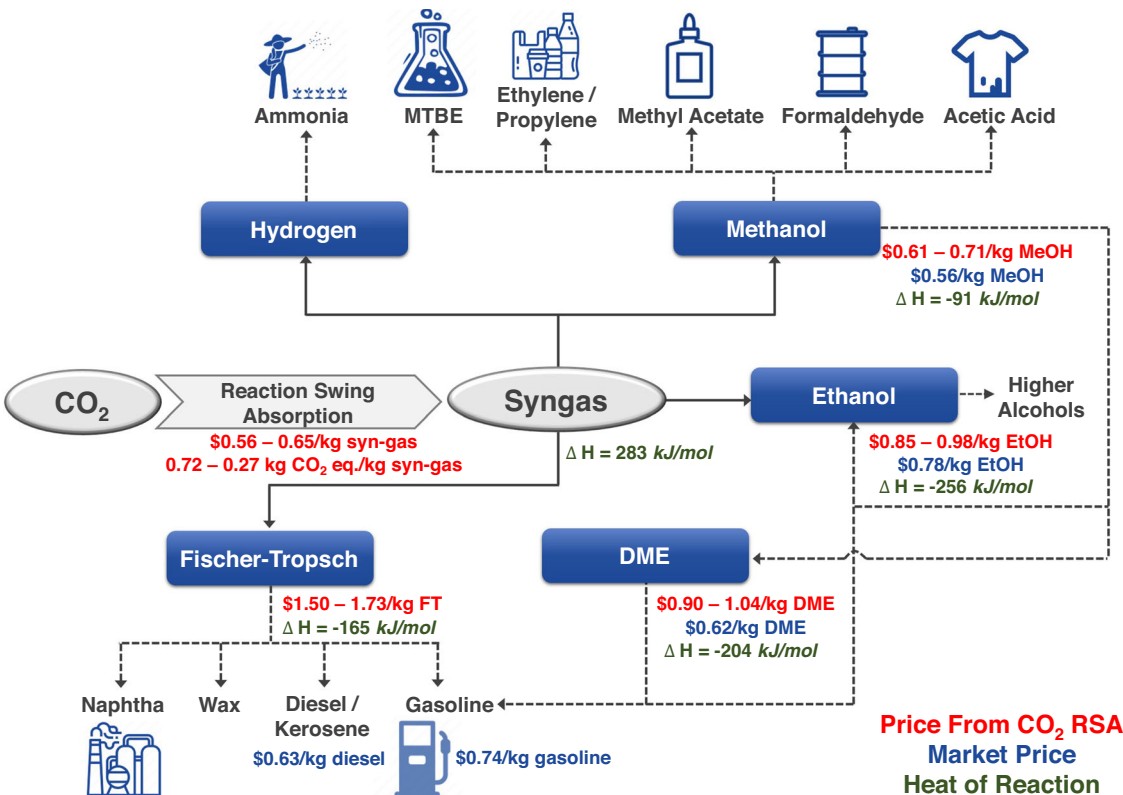

**Fig. 5 | Syngas as a versatile intermediate chemical having a competitive price when using the eco-friendly CO₂ RSA process.** Estimated levelized cost of target chemicals produced from consecutive RSA and downstream processes[39] (red), current market prices of chemicals (blue), and heat of reactions of syngas production and downstream reactions[40,41] (green). (Market prices from industries[42–44] and a report[45].

make the price highly competitive with that of fossil fuel-based syngas and achieve net-zero CO₂ emission.

## Methods

### CO₂ capture rate measurement

The CO₂ capture rate of 3 M TREA solution is measured in a pilot scale absorber. A micro gear pump (GA-V21) and a mass flow controller are used to control the simulated flue gas and absorbent flow rates. The 3–5% of simulated flue gas are prepared using 99.99% CO₂, N₂, and air. The flue gas and absorbent are count currently contacted in an absorber equipped with 3-inch Sulzer structure packing (DX) in order to maximize the mass transfer between the gas and liquid. We calculated CO₂ capture rate by measuring inlet and outlet CO₂ flow rates. To achieve the steady state condition of the absorber operation, each experiment lasted at least 45 min. More Detail information is available in Supplementary Note 1.

### Catalyst preparation and characterization

For all Ag electrodes, a carbon paper (AvCsrb MGL190) with 200 nm of Ag primary layer, which was deposited by e-beam evaporator (UL VAL Inc.) with 3 A s⁻¹ deposition rate under $10^{-6}$–$10^{-7}$ Torr of vacuum condition, was used as a substrate. The Ag nanoparticle (Ag NP) catalysts (Alfa Aesar, 99.9%) were basically deposited on the prepared substrate by spraying-coating method and modified by mixing carbon or by conducting electrochemical structural tuning. They were named according to their composition and structure such as Ag NP, Ag/C, coral-Ag, and coral-Ag/C. First, Ag NP electrode was prepared by spraying of ink solution which was composed of Ag NP (120 mg), iso-propanol (5 ml), and 5 wt.% Nafion perfluorinated resin solution (128 μl) with catalyst loading amount 1 mg cm⁻². The only difference for Ag/C electrode was that Ag NP was mixed with Ketjen black 600JD with mass ratio 75% of Ag NP in the catalyst ink solution. Coral-Ag and coral-Ag/C

electrodes were respectively prepared by oxidizing of Ag NP and Ag/C in the 0.1 M Ar-saturated KCl (Sigma-Aldrich, >99%) solution at 0.3 V (vs Ag/AgCl (3 M KCl) reference and Pt foil counter electrode for three-electrode system) for 12 h and then reducing them in 0.1 M KHCO₃ at −1.2 V (vs Ag/AgCl (3 M KCl)) for 30 min as previously reported method[32]. The morphology of the prepared Ag catalysts were characterized by scanning electron microscope (SEM) and energy dispersive X-ray spectroscopy (EDX) mapping (Hitachi Regulus 8230 with UHR cold type field emitter gun). The chemical structure of Ag electrodes was analyzed by XPS spectra using Nexsa (Thermo Fisher Scientific) with a monochromated Al-Kα (1486.6 eV) source. To understand the effect of the hydrophilicity/phobicity of the prepared Ag electrodes, contact angle was measured by a contact angle meter (DSA 25, Kruss) with 2.0 μl of DI water.

### Electrochemical measurement

All electrochemical CO₂R were performed in a commercial MEA electrolyzer (complete 5 cm² CO₂ electrolyzer, Dioxide materials). The as-prepared Ag electrode and nickel foam (200 mm length × 300 mm width × 1.6 mm thickness, MTI Korea) were used as the cathode and anode, respectively. The bipolar membrane (Fumasep FBM) was positioned between cathode and anode. The active area was controlled as 1 cm². 3 M TREA (Sigma-Aldrich, 99%) was used as the CO₂ capture solution and 1 M KOH (Sigma-Aldrich, >90%) was employed as anolyte. It is assumed that the TREA solution absorbs only CO₂ from the flue gas. Therefore, 3 M TREA solution was saturated with CO₂ for 1 h before eCO₂R. The catholyte and anolytes were respectively provided from liquid reservoirs to MEA electrolyzer and continuously circulated by using a peristaltic pump. For comparison of CO₂ captured solution, the 30 wt.% monoethanolamine or 30 wt.% diethanolamine were supplied instead of 3 M TREA. The eCO₂R of all prepared Ag electrodes were examined by chronopotentiometry using a potentiostat (VSP,

Biologic with booster 20 A) with current density range from −20, −50, −100, −150, to −200 mA cm$^{-2}$. The gas products from the eCO$_2$R were periodically quantified by online GC with a thermal conductivity detector and a flame ionization detector (Agilent 7890). MolSieve 5 A (6FT, Agilent) and Hayesep D (11FT, Agilent) packed columns were equipped, and ultrapure Ar (99.9999%) was used as the carrier gas. The products from CO$_2$ conversion were CO and H$_2$, and the total FE was confirmed to be ~100%. The FE of the products (i.e., H$_2$ and CO) was calculated by the ratio of each partial current of the products to the total current.

## Data availability
The data supporting the findings of this study are available within the article and its Supplementary Information file.

## Code availability
The codes generated in this study are available upon reasonable request to the corresponding authors.

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

## Acknowledgements

This work was supported by "Carbon to X Project" Project No. 2020M3H7A1098229 granted for K.M.G.L., H.W.L., K.P., D.K., W.J., C.W.L., H.-S.O., D.K.L., J.H.K., B.K.M., D.H.W and Project No. 2020M3H7A1098271 granted for K.T., C.K., U.L through the National Research Foundation (NRF) funded by the Ministry of Science and ICT, Republic of Korea and a KIST institutional project.

## Author contributions

K.M.G.L. and K.T. contributed to all experimental works, calculations, and manuscript preparation. C.K, and H.W.L contributed to CO$_2$ absorption experiment. K.P contributed to the analysis of captured CO$_2$ in TREA. D.K, and W.J contributed to the catalyst characterizations. C.W.L, H.-S.O, D.K.L, J.H.K, and B.K.M contributed to discussions of the system development and data analysis. D.H.W and U.L supervised this study.

## Competing interests

The authors declare no competing interests.
