## [Peer Review File · Nature Communications]

Reviewer Comments, first round -

Reviewer #1 (Remarks to the Author):

The manuscript Megagita et al. 'Toward economical application of carbon capture and utilization technology with near zero carbon emission' presents an integrated process for CO₂ capture with triethylamine and electrochemical conversion to syngas using Ag-based catalyst using a MEA type of electrolyzer. CO₂ capture and electrochemical conversion performance, techno-economic analysis (TEA), and life cycle analysis (LCA) are presented for the so-called reaction swing absorption (RSA) process. The RSA process is compared with the reverse water gas shift (RWGS) process, and the typical gas phase CO₂ electrochemical reduction process.

The manuscript is in principle interesting, but requires a very major revision before it can be accepted in any journal. The writing style should be improved, experimental results should be analyzed and discussed in more detail, TEA and LCA details are poorly described.

In revising the manuscript, the authors should consider the following:

- Please improve writing style. Check for grammar and typo's.
- On page 2, line 42: The authors mention that alkanolamines capture CO₂ as a carbamate. This is wrong, if primary and secondary amines are used then mainly carbamates are produced, but a tertiary amine results in bicarbonate.
- On page 2, line 47: the authors mention that previously used method is economically unfavorable due to the expensive supply of electrolytes. But these electrolytes are not consumed, so it needs to be purchased only once.
- In Figure 1 b, the authors plot the GWP and relative cost index for CCU (mainly methanol) processes, and conclude that CCU-based chemicals cannot compete with conventional processes. Methanol is not a good example to compare the economics of CCU processes, since it has a low market price but both thermochemical and electrochemical routes require high investment and operating costs. Furthermore, the manuscript focus on syngas, thus a comparison with methanol is anyway irrelevant. The authors should mention whether GWP100 or GWP20 is used and what negative and positive GWP means.
- On page 3, line 97: The authors mention that they propose for the first time triethylamine (TEA) ... for both CO₂ capture and bicarbonate utilization. Note that TEA has been used earlier for CO₂ capture (see Benitez-Garcia et al, Chem. Eng. Sci., 1990, 45, 3407-3415).
- On page 4, line 103: The authors mention that the theoretical absorption capacity of TEA is twice of MEA. This is true, but amines are not only selected based on the capacity, also the reactivity is important. MEA has much higher reactivity than TEA. In general, primary amines have higher reactivities than tertiary amines like TEA.
- On page 4, line 113: the authors mention that 'a smaller L/G ratio doubled the CO₂ loading, but the CO₂ absorption rate decreased by 10%'. The authors don't plot absorption rates in Figure 2, but absorption percentage. Why is Figure 2d not compared for the same CO₂ concentration? In one case 3% is used, while in the other case 5% CO₂ is used.
- Throughout the manuscript, the authors only provide CO Faradaic efficiencies. The authors should report the total gas phase composition, including FEs of hydrogen and other products. It is unclear whether the total FE of product is 100%?
- The authors mention in the manuscript that bicarbonate is the carbon source for CO₂ production. But this cannot be guaranteed from their experimental data and analysis. The authors use a bipolar membrane (BPM) that splits water into protons and hydroxides. The protons will transport towards the cathode and convert bicarbonate ions to CO₂. It is very likely that this CO₂ is converted to CO. This is one of the reasons that BPM based cell has a higher CO FE than anion or cation exchange membrane based cells (fig 3d).
- On page 6, line 170: the authors mention that the system had a stable operation for 150 h at 150 mA/cm². As can be seen in Fig 3g, the FE of CO decreases from roughly 30% to 20%, and the H₂:CO ratio changes from 2:1 to 3:1. This is problematic because typically a constant H₂:CO ratio is desired for chemical synthesis (e.g., methanol or Fischer-Tropsch). Furthermore, the electrolyzer is not a CO₂ electrolyzer, but a poor hydrogen electrolyzer, because mainly hydrogen

is produced at 4.5 V (a much higher voltage than a typical water electrolyzer). No reason to use such a electrolyzer, better electrolyzer are available for hydrogen production.

- The TEA and LCA section should be revised completely in the main text and the supporting information. More details should be provided how the authors performed the TEA and LCA. It is not straightforward to perform a TEA and LCA for relatively new non-commercial processes, in particular the processes considered by the authors. Critical assumptions and its consequences should be mentioned clearly. Presenting Aspen Plus flowsheets in the SI without a proper discussion is not enough. The authors should mention the details of the modeling. It is not trivial to model electrolyte systems, and electrochemical systems in Aspen Plus. The authors should provide more details how these processes were implemented, since Aspen does not have a standard 'electrochemical' block.

- For a proper LCA, the authors should consult: Zimmermann et al., *Front. Clim.*, 2022, 4, 841907 and references therein.

- The authors select the RWGS process and the gas phase CO₂ electrochemical conversion process for comparison. It would be more interesting to compare the RSA process with the co-SOEC process for syngas production or the Haldor-Topsoe process for CO production combined with an efficient H₂ electrolyzer.

Reviewer #2 (Remarks to the Author):

The work from Langie and co-workers propose an integrated capture and conversion system within the theme of CCU, called Reaction Swing Absorption (RSA). In RSA, CO₂ is captured from flue gas (3.5% CO₂) in the form of bicarbonate with a 3 M Triethylamine solution and then converted in a MEA electrolyzer with the help of a BPM. The system is indeed very interesting and elegant and avoids the CO₂ compression and regeneration steps which limit the valorisation of eCO₂R systems. The description is detailed, clear and of interest of an overall scientific audience. The technoeconomic analysis (TEA) is an added asset that increases the quality of the manuscript, as it is often demanded as a baseline or validation of a technological strategy. Nevertheless, I still have a major concern about the work from Langie et al., which is the lack of discussion with other competing integrated strategies such as the direct bicarbonate electrolysis from DAC solutions (i.e., using KOH as absorbent). As I mention below in my comments, the state-of-the-art (which is outdated in this manuscript) presents conversion rates overall higher compared to this study. In addition, I have other minor comments that might improve the quality of the manuscript, listed below.

Overall, the work is of interest of Nature Communications and I believe the authors can give proper answers to my concerns in a revised manuscript. Nevertheless, due to my major concern I need to revise the manuscript a second time after revisions.

Major concern:

- Line 48: The authors mention that bicarbonate electrolytes are proposed as direct-capture CO₂ conversion systems. They cite the work of Li et al. from Prof. Berlinguette's research group. Specifically, that their process showed 37% FE CO at 100 mA cm⁻². This reference is outdated. From 2019 to 2022, the same research group published two major papers on the same topic (bicarbonate electrolysis to CO/syngas), reporting, for instance:

o 50% FE CO at 100 mA cm⁻² (*ACS Energy Lett.* 2020, 5, 7, 2165–2173)

o 59% (1 atm) and 95% (4 atm) FE CO at 100 mA cm⁻² (*Energy Environ. Sci.*, 2022,15, 705-713).

This presents a huge increment in the energy efficiency of the bicarbonate electrolysis process and thus should be considered a competitor to the technology proposed in this study, where the max. FE CO at 100 mA cm⁻² is around 35% (using Coral-Ag/C). State-of-the-art references as the ones I mentioned before cannot be neglected in this study, even more when they present higher conversion rates. A TEA for bicarbonate electrolysis to be compared to the RSA is of course not needed, but the authors should clearly discuss why their approach is better than the proposed in the state-of-the-art.

Other comments:

- General comment: TEA is commonly used for triethanolamine instead of triethylamine. Triethanolamine is a common amine sorbent used in CO₂ capture research. The reaction between CO₂ and triethanolamine leads to carbamates, which is not related to the current study. I recommend the authors, to avoid confusions, use another acronym for triethylamine, such as TREA or TEYA.

- Flue gas is artificially prepared with CO₂, N₂ and O₂. However, in real flue gas other gases such as NO_x and SO_x, and solid particles are present as well. Although the amount is little, they might be significant for CCU approaches. What do the authors think about this? Have these traces an effect in the system? I understand that these traces are not added for the experiments, but the authors should mention this assumption in the Methods section.

- Proposing bicarbonate as the CO₂-supplier for eCO₂R has the advantage of avoiding amine during the capture step (and thus the formation of the undesirable C-N bond), since KOH is used as capture agent (KOH+CO₂ = KHCO₃). In fact, this method is proposed not only for flue gas capture but for Direct Air Capture, too (much lower %CO₂, 0.04%). It is true that, up to day, there are no reports of electrolysis of real absorbed KHCO₃ solutions but instead pre-fabricated KHCO₃ solutions were used as electrolyte for bicarbonate electrolysis proof-of-concept experiments (See ref. 14 from manuscript or ChemElectroChem, 9(5), e202101540 (2022)). The authors mention the necessity of using amine sorbents in addition to KOH or H₂O for generating bicarbonate from CO₂ during the capture step. What is the reasoning of this? What is the real advantage of using TEA? If I understood properly, it matters with the mechanism shown in Figure 2. But then, aren't other amines (that don't produce carbamates) able to fulfill this job? The authors should discuss the choice of amine over other potential absorbents and the use of TEA as bicarbonate formation promotor.

- Line 64: The authors claim that in RSA the release of syngas from the eCO₂R is due to the low solubility of the product in the amine solvent. Here, I have two comments:
o I agree about the thermodynamic reason (solubility), however, what about the kinetics? Isn't the release of product a limiting step? How did the authors study this? I.e., the solubility might be low, and the release of product thermodynamically favored but on the other hand the kinetics slow and therefore the eCO₂R is limited not anymore by the electrochemical reaction but in fact by the release of product.
o Shouldn't this statement be the same for other eCO₂R-to-syngas systems using carbamate as CO₂ source and amine solvents as electrolytes? Why is in RSA the low solubility of syngas a specific advantage compared to conventional carbamate electrolyzers?

- Line 100: C-NMR showed a single peak derived from bicarbonate and carbonate. However, carbonate is not mentioned anymore in the manuscript and instead all the carbon present in solution is considered bicarbonate. How were the authors sure that carbonate is not present in solution? The concentration of carbonate is relevant for the electrolysis step, as 2 protons must be released from the BPM to convert carbonate to CO₂ (while only one is needed in the case of bicarbonate).

Responses to the Comments of the Reviewer 1

The manuscript Megagita et al. ‘Toward economical application of carbon capture and utilization technology with near zero carbon emission’ presents an integrated process for CO₂ capture with triethylamine and electrochemical conversion to syngas using Ag-based catalyst using a MEA type of electrolyzer. CO₂ capture and electrochemical conversion performance, techno-economic analysis (TEA), and life cycle analysis (LCA) are presented for the so-called reaction swing absorption (RSA) process. The RSA process is compared with the reverse water gas shift (RWGS) process, and the typical gas phase CO₂ electrochemical reduction process.

The manuscript is in principle interesting, but requires a very major revision before it can be accepted in any journal. The writing style should be improved, experimental results should be analyzed and discussed in more detail, TEA and LCA details are poorly described.

In revising the manuscript, the authors should consider the following:

1. (Reviewer’s Comment) *Please improve writing style. Check for grammar and typo’s.*

(Authors’ Response) Thank you for the comment. We had checked grammar errors and typos when we prepared the original manuscript. It also had been English proofread by professional editing service, Nature Author Service. In the revised manuscript, the authors have focused more on them. If the manuscript still has any incorrect things or any parts that can be improved, please let us know.

SPRINGER NATURE
Author Services Editing Certificate

This document certifies that the manuscript
Toward economical application of carbon capture and utilization technology with
near zero carbon emission
prepared by the authors
Kezia Megagita Gerby Langie, Kyungjae Tak, Changsoo Kim, Hee Won Lee, Kwangho Park, Dongjin Kim, Wonsang Jung,
Chan Woo Lee, Hyung-suk Oh, Dong Ki Lee, Jai Hyun Koh, Byoung Koun Min, Da Hye Won, Ung Lee
was edited for proper English language, grammar, punctuation, spelling, and overall style
by one or more of the highly qualified native English speaking editors at SNAS.
This certificate was issued on **March 29, 2022** and may be verified
on the SNAS website using the verification code **F84E-1FE7-6F75-DC41-5B4P** .

Neither the research content nor the authors' intentions were altered in any way during the editing process. Documents receiving this certification should be English-ready for publication; however, the author has the ability to accept or reject our suggestions and changes. To verify the final SNAS edited version, please visit our verification page at secure.authorservices.springernature.com/certificate/verify.
If you have any questions or concerns about this edited document, please contact SNAS at support@es.springernature.com.

SNAS provides a range of editing, translation, and manuscript services for researchers and publishers around the world.
For more information about our company, services, and partner discounts, please visit authorservices.springernature.com.

2. (Reviewer's Comment) *On page 2, line 42: The authors mention that alkanolamines capture CO₂ as a carbamate. This wrong, if primary and secondary amines are used then mainly carbamates are produced, but a tertiary amine results in bicarbonate.*

(Authors' Response) We apologize for the confusion. In the original manuscript, we did not mean all alkanolamines. We tried to mean commercially available 'primary and secondary' amine among the alkanolamines, mentioning as below.

'...**in commercial CO₂-capturing absorbent** (e.g. monoethanolamine, diethanolamine, 2-amino-2-methyl-1-propanol and their mixture). **These alkanolamine** solvents capture CO₂ as a carbamate.'

To avoid further confusion, we revised the manuscript. Thank you for your valuable comment.

Changes made:

- Introduction section

Original: ... in commercial CO₂-capturing absorbent (e.g. monoethanolamine, diethanolamine, 2-amino-2-methyl-1-propanol and their mixture). These alkanolamine solvents capture CO₂ as a carbamate.

Revision: ... in commercial CO₂-capturing absorbent (e.g. monoethanolamine, diethanolamine, 2-amino-2-methyl-1-propanol and their mixture). These primary and secondary alkanolamine solvents capture CO₂ as a carbamate.

3. (Reviewer's Comment) *On page 2, line 47: the authors mention that previously used method is economically unfavorable due to the expensive supply of electrolytes. But these electrolytes are not consumed, so it needs to be purchased only once.*

(Authors' Response) Thank you for your comment, and we apologize for your confusion. We agree that common electrolytes such as metal hydroxides are not expensive in ideal situation. However, as numbers of previous studies mentioned, these metal hydroxides (e.g. KOH) can cause operational problems when they are used as electrolytes. Especially, the salt precipitation, bicarbonate deposition on electrodes, and corrosion issue of KOH electrolyte are often cited for common obstacles for the long-term operation of electrochemical eCO₂R system. Please note that KOH barely applied in aqueous CO₂ capture process with the salt formation and corrosion issues while numbers of amine-based CO₂ capture systems are industrially available. Thus, metal hydroxides should be supplied during the operation, and this additional supply can cause noticeable operating cost increase for eCO₂R system according to our previous study (*Nat. Commun.* 2019, 10, 5193). In this study, we proposed a practical method that overcomes drawback of the metal hydroxide electrolytes by applying triethylene amine.

To avoid further confusion, we revised the manuscript. Thank you for your valuable comment.

- Introduction section

Original: ... However, this method may be economically unfavorable due to the expensive supply of electrolytes...

Revision: ... However, this method may be economically unfavorable due to the expensive supply of electrolytes. Li et al. shed light on the direct-capture CO₂ conversion system by suggesting bicarbonate as a valid option for converting captured CO₂; this process showed a 37% FE for CO at -100 mA cm⁻² without the addition of supporting electrolyte, and the performance has further increased to 95% FE for CO at -100 mA cm⁻² under 4 atm pressure condition, in a recent follow-up study. In this system, KOH is considered to be a currently available CO₂ absorbent, but KOH is barely applied in aqueous CO₂ capture processes due to the salt formation and corrosion issues caused by the extremely alkaline condition. Therefore, a specific method for sustainably supplying bicarbonates from flue gas is required.

4. (Reviewer's Comment) In Figure 1 b, the authors plot the GWP and relative cost index for CCU (mainly methanol) processes, and conclude that CCU-based chemicals cannot compete with conventional processes. Methanol is not a good example to compare the economics of CCU processes, since it has a low market price but both thermochemical and electrochemical routes require high investment and operating costs. Furthermore, the manuscript focus on syngas, thus a comparison with methanol is anyway irrelevant. The authors should mention whether GWP100 or GWP20 is used and what negative and positive GWP means.

(Authors' Response) Thank you for your considerate comment. As per your comment, we added more details about the evaluation of various studies based on the relative cost index and GWP, to prevent confusion of the readers. In the main manuscript, the readers are redirected to a specific section of the Supplementary Information for a detailed explanation about the evaluation process. Also, we added more studies published recently which consider products produced via a CO-mediated pathway, to provide a more comprehensive review. Changes made to the manuscript, and additional information regarding the comments on methanol market size and GWP, are provided below:

a) Fig. 1b & Methanol as an Example

We provided additional references for syngas and diesel production from CO₂ to Fig. 1b. Regarding your concern on considering methanol as a main example, according to a review paper (*ChemSusChem* 2021, 14, 995-1015), 27% of CCU papers are related to methanol production because of the maturity of CO₂-to-methanol technology, size of the market potential, and the applicability of methanol. So, we put special attention on methanol and syngas as CO₂-based chemicals.

Figure 4. Product focus distribution within the screened papers.

b) Comparison of Syngas and Methanol

Methanol synthesis from CO₂ has two pathways from CO₂. The first path is direct CO₂ hydrogenation ($\text{CO}_2 + 3\text{H}_2 \rightarrow \text{CH}_3\text{OH} + \text{H}_2\text{O}$). The other is a two-step approach via reverse-water-gas-shift reaction ($\text{CO}_2 + \text{H}_2 \rightarrow \text{CO} + \text{H}_2\text{O}$ & $\text{CO} + 2\text{H}_2 \rightarrow \text{CH}_3\text{OH}$). The second step (CO hydrogenation) is a highly exothermic reaction ($\Delta H_{298\text{K}} = -90.77 \text{ kJ/mol}$). Since there is almost

no energy required for downstream processes after syngas production, the cost of methanol production from syngas is negligible compared to the syngas production process. Schemm et al. (*Int. J Hydrogen Energy* 2020, 45, 5395-5414) showed that raw material cost occupies 92% of the whole methanol production cost. The syngas price in our study is \$0.47/kg syngas when wind energy is used in the optimistic scenario, so the methanol price can be estimated as \$0.52/kg methanol. These two values, as well as other values for ethanol, DME, and Fischer-Tropsch, can be found in Fig. 5.

c) GWP

The GWP value is calculated differently for various studies, according to the range of the system boundary designated within the study. When a full system boundary is implemented, where the CO₂ source is included along with the CCU process, the GWP values are calculated by subtracting the conventional GWP value from the GWP of the proposed process. For a restricted system boundary, where only the proposed process is selected as the system of analysis, neglecting the CO₂ source, the GWP value provided within the study is used.

All of the GWP values provided within the references are regarded as GWP100 values, which indicates the GWP within a 100-year period. While some studies state the use of GWP100 during LCA, most of the studies do not specify the LCA evaluation method of use or the GWP evaluation period. Since the most widely used LCA evaluation method is the ReCiPe 2016 Hierarchy method, which evaluates the GWP100 of various chemicals, we evaluated our study based on GWP100. This can be confirmed from a report on ReCiPe 2016 v1.1 (https://pre-sustainability.com/legacy/download/Report_ReCiPe_2017.pdf):

Table 2.1. Value choices in the modelling of the effect of GHGs

Choice category	Individualist	Hierarchist	Egalitarian
Time horizon	20 years	100 years	1,000 years
Climate-carbon feedbacks included for non-CO ₂ GHGs	No	Yes	No ¹
Future socio-economic developments	Optimistic	Baseline	Pessimistic
Adaptation potential	Adaptive	Controlling	Comprehensive

¹ Ideally, Climate-Carbon feedbacks should be included for this perspective; however GWPs including Climate Carbon feedbacks are not available for a 1,000-year time horizon.

These criteria have been added to the Supplementary Text 4.3.3 to help understand the contents of Figure 1(b).

Changes made:

- Figure 1b

Original:

Revision:

- On page 2, line 86,

Revision (added): Note that GWP is presented as kgCO₂ eq. per functional unit(FU). Details regarding the meta-analysis is presented in the Supplementary Note 4.3.3.

- On page 28 of supplementary information

Revision:

Regarding the meta-analysis presented in Fig. 1(b), the relative cost indices and the global warming potential (GWP) of the different studies are evaluated based on the following criteria: first, the $C_{\text{ProductCCU}}$ value is determined as the base case cost value provided by the specific literature. Most studies provide cost values for various scenarios, such as changes in renewable electricity costs, and policy changes such as carbon credits. These changes can greatly alter the $C_{\text{ProductCCU}}$ values, but inherit large uncertainties in terms of the implementation period, or the level of implementation. To allow a fair comparison, the base case of the proposed process/system, which is based on a realistic scenario, is taken as the $C_{\text{ProductCCU}}$ value of that study. Secondly, the GWP value is calculated differently according to the range of the system boundary designated within the study. When a full system boundary is implemented, where the CO_2 source is included along with the CCU process, the GWP values are calculated by subtracting the conventional GWP value from the GWP of the proposed process. For a restricted system boundary, where only the proposed process is selected as the system of analysis, neglecting the CO_2 source, the GWP value provided within the study is used. Thirdly, all of the GWP values provided within the references are regarded as GWP100 values, which indicates the GWP within a 100-year period. While some studies state the use of GWP100 during LCA, most of the studies do not specify the LCA evaluation method of use or the GWP evaluation period. Since the most widely used LCA evaluation method is the ReCiPe 2016 Hierarchy method, which evaluates the GWP100 of various chemicals, it is assumed that all of the studies are evaluated based on this method.

5. (Reviewer's Comment) *On page 3, line 97: The authors mention that the propose for the first time triethylamine (TEA) ... for both CO₂ capture and bicarbonate utilization. Note that TEA has been used earlier for CO₂ capture (see Benitez-Garcia et al, Chem. Eng. Sci., 1990, 45, 3407-3415).*

(Authors' Response) Thank you for the comment. According to your comment, we eliminated expression "for the first time". We insisted in the original manuscript that it is the first use of triethylamine (TEA) not only for CO₂ capture but also for direct conversion of amine-captured CO₂. Of course, we know numbers of amines have been investigated for CO₂ capture, including TEA. The paper of Benitez-Garcia et al. is one of these papers, only dealing with very basic level of CO₂ absorption experiments and its thermodynamic modeling. We revised the manuscript for clarity of the meaning.

Changes made:

- On page 3, line 98

Original: Thus, it is important to find a new amine that captures CO₂ in a mild form, such as bicarbonate. After screening various amines, including primary, secondary and tertiary amines, we propose for the first time that triethylamine (TEA), which contains aliphatic groups but not hydroxyl groups, is an ideal solvent for both CO₂ capture and bicarbonate utilization.

Revision: Thus, it is important to find a new amine that captures CO₂ in a mild form such as bicarbonate, and directly converts amine-captured CO₂. After screening various amines, including primary, secondary, and tertiary amines, we found that triethylamine (TEA) is an ideal solvent for both CO₂ capture and bicarbonate utilization, which contains aliphatic groups but not hydroxyl group.

6. (Reviewer's Comment) *On page 4, line 103: The authors mention that the theoretical absorption capacity of TEA is twice of MEA. This is true, but amines are not only selected based on the capacity, also the reactivity is important. MEA has much higher reactivity than TEA. In general, primary amines have higher reactivities than tertiary amines like TEA.*

(Authors' Response) Thank you for your valuable comment and we completely agree with that amine selection should not be based solely on absorption capacity. There are many criteria for amine selection, such as absorption capacity, reactivity, heat of regeneration, stability, and yield of bicarbonate. We selected triethylamine (TREA) due to its high yield of bicarbonate.

In the case of reactivity, the absorption experiment showed that TREA has enough absorption performance (Figs. 2 and S2), although MEA has a better absorption performance (please note results in Table S2 for MEA and TREA column size).

Changes made:

- On page 4, line 108

Original: Note that the theoretical absorption capacity of the TEA is twice that of monoethanolamine.

Revision: Note that the theoretical absorption capacity of the TREA is twice that of monoethanolamine, although monoethanolamine has higher reactivity.

7. (Reviewer's Comment) *On page 4, line 113: the authors mention that 'a smaller L/G ratio doubled the CO₂ loading, but the CO₂ absorption rate decreased by 10%'. The authors don't plot absorption rates in Figure 2, but absorption percentage. Why is Figure 2d not compared for the same CO₂ concentration? In one case 3% is used, while in the other case 5% CO₂ is used.*

(Authors' Response) Thank you for the comment. We changed Figure 2 in the revised manuscript with additional experiments of CO₂ absorption. For clarity and understandability, we added CO₂ absorption capacity (mol CO₂/mol TREA).

There are three control variables: liquid flow, gas flow, and CO₂ concentration in gas. Fig. 2c shows the effect of liquid flow on CO₂ absorption rate (or CO₂ removal rate) as well as CO₂ absorption capacity (mol CO₂/mol TREA) under fixed gas flow rate (0.8 m³/h) and CO₂ concentration in gas feed (3% CO₂).

In Fig. 2d in the revised manuscript, CO₂ absorption capacity is plotted based on L/CO₂ ratio and shows a linearly decreasing trend, regardless of flow rates (or $\propto 1/\text{space time}$) of liquid and gas in the absorber and CO₂ concentration in gas.

Changes made:

- On page 4, line 109

Original: The CO₂ absorption capacity of TEA solvent was measured with a bench scale absorption column equipped with structure packing (Fig. 2b and S2). We measured the CO₂ absorption rates at liquid/gas ratios (L/G, L m⁻³) ranging from 3.75 to 6.25 and with synthesized flue gas containing 3% CO₂. Fig. 2c shows that the CO₂ absorption rate gradually increased with increases in the L/G ratio. This result also indicated that at least 84% of the CO₂ can be removed from flue gas even if the L/G ratio is reduced to 3.75. The maximum CO₂ absorption rate obtained in this study was approximately 96.7% (L/G =6), and this is far higher value than those of conventional CO₂ capture processes. The CO₂ loading (mol CO₂/mol TEA) for the absorption experiment with 3% CO₂ was relatively small (0.1 ~ 0.12) compared to those reported in previous studies, because of the low CO₂ concentration and the short column height. Fig. 2d presents the effect of changing the L/G ratio on the CO₂ loading and absorption performance. A smaller L/G ratio doubled the CO₂ loading, but the CO₂ absorption rate decreased by approximately 10%.

Revision: The CO₂ absorption capacity of TREA solvent was measured with a bench scale absorption column equipped with structure packing (Fig. 2b and S2). We measured the CO₂ absorption rates and capacities at liquid feed/gas CO₂ feed (L/CO₂) ratios ranging from 0.07 to 0.21 with synthesized flue gas containing 3–5% CO₂. Fig. 2c shows that the higher the L/CO₂ ratio at fixed gas flow and CO₂ concentration, the greater the CO₂ absorption rate but the less CO₂ absorption capacity. Fig. 2d presents a linear relationship between L/CO₂ ratio and CO₂ absorption capacity, regardless of liquid flow (2–5 L/h), gas flow (0.5–0.8 m³/h), and CO₂ concentration (3–5%).

- Figure 2c and 2d

Original:

(c) CO₂ absorption experiment results with varying TEA flow rates. Minimum absorption performance is 84% with TEA flow rate of 2 L h⁻¹ (L/G ratio 3.75). (d) Relative effect of varying L/G ratios on CO₂ loading and absorption performance of TEA. With smaller L/G ratios, the CO₂ loading is increased by factors of 2 (TEA flow rate of 2 L h⁻¹, the orange bar represents the result for flue gas containing 5% CO₂) and 1.6 (TEA flow rate of 3 L h⁻¹), while CO₂ absorption is decreased only by 8.7% and 12.4%, respectively.

Revision:

(c) CO₂ absorption experiment results with varying TREA flow rates at fixed gas flow rate (0.8 m³/h) and CO₂ concentration in gas flow (3% CO₂). (d) Effect of L/CO₂ ratios on CO₂ absorption capacity at various values of TREA flow rates (2–5 L/h), gas flow rates (0.5–0.8 m³/h), and CO₂ concentrations (3–5%). CO₂ absorption capacity shows a linear correlation with L/CO₂ ratio, regardless of TREA flow rate, gas flow rate, and CO₂ concentration.

- Figure S2

Original:

Amine Solvent [L/hr]	Flue Gas [m ³ /hr]	L/G [-]	Inlet CO ₂ [%]	Outlet CO ₂ [%]	CO ₂ Absorption [%]
5	0.8	6.25	3.21	0.28	91.15
4	0.8	5.00	3.06	0.40	87.06
3	0.8	3.75	3.06	0.48	84.48
3	0.5	6.00	2.95	0.09	96.86
2	0.6	3.33	5.00	0.68	86.40
2	0.5	4.00	2.95	0.15	95.08

Revision:

Amine Solvent [L/hr]	Flue Gas [m ³ /hr]	Inlet CO ₂ [%]	Outlet CO ₂ [%]	L/CO ₂ [-]	CO ₂ Absorption Rate [%]	CO ₂ Absorption Capacity [mol CO ₂ /mol TREA]
2	0.5	2.95	0.15	0.136	95.1	0.104
2	0.6	5.00	0.68	0.067	86.4	0.193
3	0.8	3.06	0.48	0.123	84.5	0.103
3	0.5	2.95	0.09	0.203	96.9	0.071
4	0.8	3.06	0.40	0.163	87.1	0.079
5	0.8	4.85	0.46	0.129	90.5	0.105
5	0.8	4.40	0.43	0.142	90.2	0.094
5	0.8	3.21	0.28	0.195	91.2	0.070

8. (Reviewer's Comment) Throughout the manuscript, the authors only provide CO Faradaic efficiencies. The authors should report the total gas phase composition, including FEs of hydrogen and other products. It is unclear whether the total FE of product is 100%?

(Authors' Response) As we mentioned in methods in the manuscript, total Faradaic efficiencies of CO and H₂ are nearly 100% for all experiments as shown in below Figure. This indicates that CO and H₂ are the only products during eCO₂R in this system. We newly inserted the several graphs that show Faradaic efficiencies of H₂ in the revised Supporting information as reviewer's comment.

Changes made:

- Newly inserted Figure S3 in the revised supplementary information

Figure S3. CO FE and H₂ FE for different catalysts with various membranes. (a) CO FEs (plain) and H₂ FEs (dash) for Ag/C measured with various applied current densities in monoethanolamine (black), diethanolamine (red), and TREA (blue). (b) CO FEs (plain) and H₂ FEs (dash) for Ag NP (black), Ag/C (yellow), coral-Ag (red), and coral-Ag/C (blue) in 3 M TREA. (c) CO FEs (plain) and H₂ FEs (dash) for Ag/C measured with various membranes, including an anion exchange membrane (black), cation exchange membrane (red) and bipolar membrane (blue), in 3 M TREA.

9. (Reviewer's Comment) *The authors mention in the manuscript that bicarbonate is the carbon source for CO₂ production. But this cannot be guaranteed from their experimental data and analysis. The authors use a bipolar membrane (BPM) that splits water into protons and hydroxides. The protons will transport towards the cathode and convert bicarbonate ions to CO₂. It is very likely that this CO₂ is converted to CO. This is one of the reasons that BPM based cell has a higher CO FE than anion or cation exchange membrane based cells (fig 3d).*

(Authors' Response) We are quite on the same pages as reviewer's comment. As reviewer's point out, we also expected that the bicarbonate initially converted to CO₂ by protons at near the membrane, and then this CO₂ is catalytically reduced to CO at the surface of cathode. This reaction mechanism was already inserted in original version of manuscript as Figure 3b. Especially, since BPM splits water into protons and hydroxides, which respectively supply to cathode and anode, we evaluated that this unique operation of BPM contributes to sustainable eCO₂R reaction in TREA solution. Meanwhile, anion exchange membrane (AEM) and cation exchange membrane (CEM) applied systems show negligible performances due to relatively unfavorable reaction conditions for bicarbonate-to-CO₂ and CO₂-to-CO compared to BPM applied system. In case of AEM, the amount of produced CO₂ from bicarbonate is limited because proton is barely supplied from membranes. Furthermore, bicarbonate at the cathode part can crossover to anode part due to its anion exchange ability, which eventually degrades the eCO₂R performance and system sustainability. This mechanism referred to the previous report (Joule, 2019, 3, 1487) which utilizes the 3M KHCO₃ as a source for CO₂. In this study, the control experiments reveal that CO₂ production and CO₂-to-CO conversion in KHCO₃ was more facilitated by BPM due to higher concentration of proton donors provided by the BPM than the AEM. In case of CEM, the protons can be supplied through the membrane so that CO₂ can be produced from bicarbonate better than AEM. However, CEM can crossover to K⁺ from anode part to cathode part which still cause the problem to separate the electrolytes. As results, BPM can be best and only option for eCO₂R in direct conversion of captured CO₂ in TREA solution.

10. (Reviewer's Comment) On page 6, line 170: the authors mention that the system had a stable operation for 150 h at 150 mA/cm². As can be seen in Fig 3g, the FE of CO decreases from roughly 30% to 20%, and the H₂:CO ratio changes from 2:1 to 3:1. This is problematic because typically a constant H₂:CO ratio is desired for chemical synthesis (e.g., methanol or Fischer-Tropsch). Furthermore, the electrolyzer is not a CO₂ electrolyzer, but a poor hydrogen electrolyzer, because mainly hydrogen is produced at 4.5 V (a much higher voltage than a typical water electrolyzer). No reason to use such an electrolyzer, better electrolyzers are available for hydrogen production.

(Authors' Response) Thank you for the comments. Those three comments are answered below, respectively.

a) Stability & H₂/CO ratio

We assumed that the slightly degraded performance during long-term operation might be derived from the lab-scale experiment limitation such as i) the possibility of unbalancing between CO₂ capture and bicarbonate consumption in the lab-scale CO₂ absorption reactor and electrolyzer, and ii) the decrease TREA concentration in electrolyte due to vaporization of TREA. Thus, we re-tried the stability test at -100 mA cm⁻² condition, which can reduce the rate of bicarbonate consumption by lowering the current density, and by using air-tight absorption reactor. Finally, the performance is stable until 70 h with well-maintained potential and CO and H₂ FEs, which indicated the maintained H₂:CO ratio. We revised the main figure with this new data.

Figure. (a) Previous long-term CO production represented as the H₂:CO ratio (black), CO FEs (red), and cell voltages (blue) for coral-Ag/C at -150 mA cm⁻² in 3 M TREA. (b) New long-term eCO₂R performance represented as CO FEs (red), H₂ FEs (black) and cell voltages (blue) for coral-Ag/C at -100 mA cm⁻² in 3 M TREA.

Changes made:

- Replacement of Figure 3g in revised manuscript
- In manuscript line 170-173

Original: . This optimized system configuration (coral-Ag/C cathode and bipolar membrane) exhibited stable long-term operation.

Revision: . This optimized system configuration (coral-Ag/C cathode and bipolar membrane) exhibited stable performance with 35% of CO FE, during a 70 h operation of chronopotentiometry experiment at -100mA cm^{-2} (Fig. 3g). During chronopotentiometry experiment at -100 mA cm^{-2} , the RSA system demonstrated stable performance with 35% of CO FE.

b) High voltage

This study used the electrolyzer for co-production of CO and H₂. We agree that the cell voltage of 4.5V is high. In our model, we used 3.51V in order to meet the 2:1 hydrogen and CO ratio. The optimum voltage we found based on the experimental result is also much lower than 4.5V. Also, we considered two options for H₂ supply: producing H₂ by this electrolyzer and purchasing H₂ from water electrolysis (PEM Electrolyzer for 1,500 kg-H₂/day. The cost of H₂ from water electrolysis was based on a DOE report (Final Report - Hydrogen Production Pathways Cost Analysis (2013-2016)) cited in the Supplementary Information.

In the energy mix case, the actual cost-optimal cell voltage in the RSA is much lower (3.51V for current and 2.49V for optimistic). The detailed comparison can be seen in the table below. The additional cost is based on the figure below, which consists of the CSD (Compression, Storage, and Dispensing) cost and all production cost except electricity cost (electricity cost is calculated based on electricity price in each energy case with electricity use data provided in the literature). For both current and optimistic scenarios, the CO₂ electrolyzer is a better option for H₂ supply.

	By-product of CO ₂ electrolyzer		Water Electrolysis	
	Current	Optimistic	Current	Optimistic
Voltage (V)	3.51	2.49		
CO FE (%)	38.77	39.11		
Electricity Use (kWh/kg H ₂)	93.20	66.24	54.30	50.20
Electricity Price (\$/kWh)	0.07	0.04	0.07	0.04
Electricity Cost (\$/kg H ₂)	6.37	2.86	3.71	2.16
Additional Cost (\$/kg H ₂)			4.18	2.31
Total Cost (\$/kg H ₂)	6.37	2.86	7.89	4.47

¹ DOE report (Final Report - Hydrogen Production Pathways Cost Analysis (2013-2016))

11. (Reviewer's Comment) *The TEA and LCA section should be revised completely in the main text and the supporting information. More details should be provided how the authors performed the TEA and LCA. It is not straightforward to perform a TEA and LCA for relatively new non-commercial processes, in particular the processes considered by the authors. Critical assumptions and its consequences should be mentioned clearly. Presenting Aspen Plus flowsheets in the SI without a proper discussion is not enough. The authors should mention the details of the modeling. It is not trivial to model electrolyte systems, and electrochemical systems in Aspen Plus. The authors should provide more details how these processes were implemented, since Aspen does not have a standard 'electrochemical' block.*

(Authors' Response) Thank you for the comment. The TEA and LCA parts are totally revised. Please note that we use the same process model from our previous study (*Nat. Commun.* 2019, 10, 5193) and we did not include the same detail to avoid duplicated explanation. In code availability section, we also mentioned that source code is available in our homepage. We also include a sentence in SI indicating that more details and code are available in our previous publication.

Changes made:

- On page SI, Page S18

Revision: Note that more detail description regarding modeling and flowsheeting algorithms are available in Na et al.

a) Aspen Plus process simulation

The detailed information and explanations for Aspen Plus simulations are added in the SI, such as what chemisorption reactions are, how reactions in electrolyzers and RWGS reactor are modeled, which thermodynamic property package is used in the simulation (ELECNRTL) and how components are added (databanks in Aspen Plus is used for all components, except TREA and TREA⁺ which are added by the user-defined molecular structures). Also, Table S2 has more information for the modeling conditions in Aspen Plus. Some missing information can be found in the literature (referred in the manuscript, for example, "The RWGS process is modeled, based on the result of the stable 80-hour operation of RWGS reaction from Sun et al. [2]"). Therefore, almost all information is provided in the manuscript, and the missing information (if any) are very minor ones that have negligible impacts.

b) TEA

In the case of capital investment cost calculations for TEA, the SI shows the detailed methodology with equations. These are based on a textbook (ref. 7 in the SI). The parameters for equipment cost calculations are seen in the table below, which is not added in the manuscript due to the redundancy. The parameters for PSA and electrolyzers can be found from papers in the literature (ref. 5 in the SI). In the revised SI, it is added how CAPEX is calculated after obtaining equipment costs.

TABLE 4.12 Base Costs for Process Equipment

Equipment Type	C_0 (\$10 ³)	S_0	Range(\$)	α	MF2/MF4/MF6/MF8/MF10
Process furnaces $S = \text{Absorbed duty (10}^9\text{Btu/hr)}$	100	30	10–300	0.83	2.27/2.19/2.16/2.15/2.13
Direct fired heaters $S = \text{Absorbed duty (10}^9\text{Btu/hr)}$	20	5	1–40	0.77	2.23/2.15/2.13/2.12/2.10
Heat exchanger Shell and tube, $S = \text{Area (ft}^2\text{)}$	5	400	100–10 ⁴	0.65	3.29/3.18/3.14/3.12/3.09
Heat exchanger Shell and tube, $S = \text{Area (ft}^2\text{)}$	0.3	5.5	2–100	0.024	1.83/1.83/1.83/1.83/1.83
Air coolers $S = [\text{Calculated area (ft}^2\text{)}]/15.5]$	3	200	100–10 ⁴	0.82	2.31/2.21/2.18/2.16/2.15
Centrifugal pumps	0.39 0.65 1.5	10 $2 \cdot 10^3$ $2 \cdot 10^4$	$10^{-2} \cdot 10^3$ $2 \cdot 10^3 - 2 \cdot 10^4$ $2 \cdot 10^4 - 2 \cdot 10^5$	0.17 0.36 0.64	3.38/3.28/3.24/3.23/3.20 3.38/3.28/3.24/3.23/3.20 3.38/3.28/3.24/3.23/3.20
$S = \text{C/H factor (gpm} \times \text{psi)}$					
Compressors $S = \text{brake horsepower}$	23	100	30–10 ⁴	0.77	3.11/3.01/2.97/2.96/2.93
Refrigeration $S = \text{ton refrigeration (12,000 Btu/hr removed)}$	60	200	50–3000	0.70	1.42

(Data from Guthrie, 1969)

For operating cost calculations for TEA, the components and their proportion are based on another textbook (ref. 8 in the SI). Fig. S15 is the edited version of Table 6-17 in ref. 8. The other parameters used for OPEX are in Table S3, such as raw material price, utility price, depreciation method, and interest rate.

Table 6-17 Estimation of capital investment cost (showing individual components)

The percentages indicated in the following summary of the various costs constituting the capital investment are approximations applicable to ordinary chemical processing plants. It should be realized that the values given vary depending on many factors, such as plant location, type of process, and complexity of instrumentation.

- I. **Direct costs** = material and labor involved in actual installation of complete facility (65–85% of fixed-capital investment)
 - A. Equipment + installation + instrumentation + piping + electrical + insulation + painting (50–60% of fixed-capital investment)
 1. Purchased equipment (15–40% of fixed-capital investment)
 2. Installation, including insulation and painting (25–55% of purchased-equipment cost)
 3. Instrumentation and controls, installed (8–50% of purchased-equipment cost)
 4. Piping, installed (10–80% of purchased-equipment cost)
 5. Electrical, installed (10–40% of purchased-equipment cost)
 - B. Buildings, process, and auxiliary (10–70% of purchased-equipment cost)
 - C. Service facilities and yard improvements (40–100% of purchased-equipment cost)
 - D. Land (1–2% of fixed-capital investment or 4–8% of purchased-equipment cost)
- II. **Indirect costs** = expenses which are not directly involved with material and labor of actual installation of complete facility (15–35% of fixed-capital investment)
 - A. Engineering and supervision (5–30% of direct costs)
 - B. Legal expenses (1–3% of fixed-capital investment)
 - C. Construction expense and contractor's fee (10–20% of fixed-capital investment)
 - D. Contingency (5–15% of fixed-capital investment)
- III. **Fixed-capital investment** = direct costs + indirect costs
- IV. **Working capital** (10–20% of total capital investment)
- V. **Total capital investment** = fixed-capital investment + working capital

c) LCA

We mentioned that i) an LCA tool, SimaPro, is used, ii) ReCiPe 2016(H) method is employed, iii) 'Cradle-to-Gate' approach is adopted, and iv) ecoinvent database is used. Also, Fig. 4 shows the system boundaries for LCAs.

LCAs were conducted by Matlab. Matlab gets process simulation results from Aspen Plus to gather life cycle inventory. Then, life cycle impact parameters extracted from SimaPro are used for LCA calculations. The required parameters are materials and energy related ones as well as transportation and chemical plant related ones. All these parameters without transportation related ones are extracted from SimaPro. Because syngas is produced at a site near coal-fired power plant, there is no transportation, except external H₂ and amines for make-up. We ignored this. All these things are explained in the SI.

d) TEA and LCA for non-commercial processes

Although all three processes in this system are not commercialization, most of equipment in those processes are conventional ones: pumps, compressors, columns, coolers, process heaters, heat exchangers, flash drums, etc. Even for the RWGS reactor, it is a widely used fixed bed reactor. Therefore, equipment costs for those can be calculated by the data in year of 1968 (ref. 5 in the SI). The new ones are electrolyzer and PSA only, which are calculated based on a recent paper (ref. 5 in the SI).

In the case of LCA, non-commercial processes are usually assessed by ‘Cradle-to-Gate’ methods in the literature.

The processes are non-commercial, have low TRL levels, so are highly uncertain. To overcome this, we conducted sensitivity analyses using 5,000 samples. Therefore, readers can evaluate those processes from the GSA results.

12. (Reviewer’s Comment) For a proper LCA, the authors should consult: Zimmermann et al., *Front. Clim.*, 2022, 4, 841907 and references therein.

(Authors’ Response) Thank you for the valuable suggestion with the guidance paper.

According to the paper (Zimmermann et al., *Front. Clm.* 2022, 4, 841907), many things should be considered when conducting LCAs at early development stage (low TRL level). For example, lack of a well-defined function or market. We used 1 kg of syngas as a functional unit, based on the recommendation of their previous paper (Muller et al., *Front. Energy Res.*, 2020, 8, 15). Another example is data availability, missing data, and uncertainty. We conducted GSAs for TEA and LCA. They (Muller et al., *Front. Energy Res.*, 2020, 8, 15) also recommended a ‘cradle-to-gate’ approach for our case, and we employed this approach.

Considering other CCU-related papers, many LCA studies have been conducted for comparative assessment using ‘Cradle-to-Gate’ with mass functional unit (See table below, reported in Thonemann, *Appl. Energy* 2020, 263, 114599). Our LCA study is in accordance with these studies, where we compared three processes using ‘Cradle-to-Gate’ under the basis of mass of produced syngas.

Table 1
Methodological differences within the analyzed studies.

Authors	Aim	Geographic reference area	Temporal coverage	System boundary	Functional unit		Handling of multi-functionality					
					related to	based on	Subdivision	System expansion	Substitution	Allocation		
							Physical	Mass	Economic			
Ahn et al. (2019) [53]	Comparative assessment	KR	Today	Cradle-to-gate	Mass	Output						
Aldaco et al. (2019) [60]	Comparative dynamic assessment	EU	Today (2015) – 2040 (5 year rhythm)	Cradle-to-gate	Mass	Output						X
Aresta et al. (2002) [35]	Comparative assessment	EU	Today	Cradle-to-gate	Mass	Output						X
Aresta & Galatola (1999) [34]	Comparative assessment	EU	Today	Cradle-to-gate	Mass	Output						X
Biermacki et al. (2018) [42]	Comparative assessment	DE	Today	Cradle-to-gate	Mass	Output			X			X
Büttner et al. (2019) [63]	Comparative assessment	n. s.	Today	Cradle-to-gate	Mass	Output			n. s.			
Collet et al. (2016) [45]	Comparative assessment	FR	Today, 2020, 2030, 2050	Cradle-to-gate	Energy	Output			X			
Deutz et al. (2018) [44]	Comparative assessment	EU	Today, 2020, 2050	Cradle-to-gate	Mass	Output			n. s.			
Dominguez-Ramos & Irabien et al. (2019) [64]	Environmental impact assessment	n. s.	Today, 2050	Cradle-to-gate	Distance	Output			X			
Dominguez-Ramos et al. (2015) [65]	Comparative assessment	n. s.	Today, long-term	Cradle-to-gate	Mass	Output			X			
Durkin et al. (2019) [51]	Comparative assessment Identify hotspots	BR, ES	Today	Cradle-to-gate	Mass	Output			X	X	X	X
Falter et al. (2016) [48]	Guiding process engineers Environmental impact assessment	n. s.	Today	Cradle-to-gate	Volume	Output				X		
Fernández-Dacosta et al. (2019) [66]	Comparative assessment	NL	Today	Cradle-to-gate	Energy	Output			X			
Fernández-Dacosta et al. (2018) [67]	Comparative assessment	NW EU	Today (2016)	Cradle-to-gate	Mass & energy	Output			X			
Fernández-Dacosta et al. (2017) [68]	Comparative assessment	NW EU	Today (2015)	Cradle-to-gate	Mass & energy	Output			X			
García-Herrero et al. (2016) [69]	Comparative assessment Guiding process engineers	ES	Today	Cradle-to-gate	Mass	Output			n. s.			
Griffiths et al. (2013) [70]	Comparative assessment	UK	Today	Cradle-to-gate	Mass	Output					X	
Hoppe & Brinquez (2016) [71]	Comparative assessment	DE	Today, 2030, 2050	Cradle-to-gate	Mass	Output						X
Hoppe et al. (2016) [72]	Comparative assessment	DE	Today	Cradle-to-gate	Mass	Output						X
Hoppe et al. (2018) [73]	Comparative assessment	DE	Today	Cradle-to-gate	Mass	Output						X
Jens et al. (2019) [74]	Comparative assessment	DE	Today	Cradle-to-gate	Mass	Output			X			
Kim et al. (2011) [75]	Comparative assessment	US	Today	Cradle-to-gate	Mass	Output			n. s.			
Koj et al. (2018) [76]	Comparative assessment	DE	2050	Cradle-to-gate	Distance	Output					X	
Kongpanna et al. (2015) [77]	Comparative assessment	n. s.	Today	Gate-to-gate	Mass	Output			n. s.			
Matzen & Demirel (2016) [78]	Environmental impact assessment	n. s.	Today	Cradle-to-gate	Mass	Output						X
Meunier et al. (2019) [79]	Environmental impact assessment	EU	Today	Gate-to-gate	Mass	Output						X
Meys et al. (2019) [80]	Comparative assessment	DE	Today	Cradle-to-gate	Mass	Output			X			
Parra et al. (2017) [81]	Comparative assessment	CH, EU	Today (2015–2025)	Cradle-to-gate	Energy	Output						X
Reiter & Lindorfer et al. (2015) [82]	Comparative assessment Identify hotspots	EU	Today	Cradle-to-gate	Energy	Output			n. s.			
Rumayor et al. (2018) [55]	Comparative assessment	n. s.	Today	Cradle-to-gate	Mass	Output						X
Rumayor et al. (2019a) [56]	Comparative assessment	ES	Today	Cradle-to-gate	Mass	Output			n. s.			
Rumayor et al. (2019b) [57]	Comparative assessment	n. s.	Today	Cradle-to-gate	Mass	Output						X
Schakel et al. (2016) [83]	Comparative assessment	NW EU	Today (2015)	Cradle-to-gate	Energy	Output			X			
Sternberg & Bardow (2015) [49]	Comparative assessment	US, BR, DE, JP	Today, 2020	Cradle-to-gate	Energy	Input				X		
Sternberg & Bardow (2016) [46]	Comparative assessment	DE	Today, 2020, 2030	Cradle-to-gate	Mass & energy	Output			X			

(continued on next page)

Changes made:

- On Page 20 of Supplementary Information

Original: LCAs with parameter values obtained from SimaPro V9.3 are conducted by Matlab.

The ReCiPe 2016 is used as a 'Cradle-to-Gate' method with the ecoinvent database V3.8 for the system boundaries in Fig. 4c.

Revision: LCAs with impact parameter values of materials, energies, etc. obtained from SimaPro V9.3 are conducted by Matlab, based on the process simulation results from Aspen Plus. The ReCiPe 2016(H) is used as a 'Cradle-to-Gate' method with the ecoinvent database V3.8 for the system boundaries in Fig. 4c. For simplicity, transportation of raw materials (external H₂ and amines) is ignored for LCA. Because the three processes in this study have early technology maturity with different technology readiness levels, they are highly uncertain and lack data availability [9–10]. Many LCA studies of CCU technology in the literature have been conducted for comparative assessment using 'Cradle-to-Gate' approach with functional unit of mass [11]. Accordingly, this study employed the same methodology and conducted sensitivity analysis and scenario analysis.

13. (Reviewer's Comment) *The authors select the RWGS process and the gas phase CO₂ electrochemical conversion process for comparison. It would be more interesting to compare the RSA process with the co-SOEC process for syngas production or the Haldor-Topsoe process for CO production combined with an efficient H₂ electrolyzer.*

(Authors' Response) Thank you for the comment and we actually considered the SOEC process for syngas production as one of alternative options for replacing thermochemical synthesis (RWGS).

Our purpose is to find a net-zero technology with economic feasibility. However, the SOEC is operated at high temperatures (600–850°C). This requires a large amount of heat which emits a lot of CO₂. While electrochemical CO₂ reduction has high potential for reducing CO₂ emissions in the future since renewable energy can be implemented, high temperature operating conditions can only be achieved with fossil fuels, which makes SOEC a short-term CCU application. Also, many electrolyzers in SOEC are using ceramic materials, such as YSZ, leading to large capital investment. As a result, we ruled out the SOEC process for the study.

Changes made:

- In page 12 of supplementary information

Original: The three CCU processes considered in this study are thermal CO₂ conversion (RWGS, reverse water gas shift reaction), electrochemical CO₂ conversion (gas eCO₂R), and electrochemical bicarbonate conversion (RSA, reaction swing absorption) for syngas production.

Revision: The purpose of modeling in this study is to find an economically and environmentally feasible CCU technology capable of replacing the thermochemical CO₂ conversion technology. The three CCU processes considered in this study are thermal CO₂ conversion (RWGS, reverse water gas shift reaction), electrochemical CO₂ conversion (gas eCO₂R), and electrochemical bicarbonate conversion (RSA, reaction swing absorption) for syngas production. Solid oxide electrolysis cell (SOEC) can also be an option for syngas production, but SOEC requires a large amount of heat to be operated at high temperatures (600–850°C) that causes a lot of CO₂ emission. Consequently, this study did not consider SOEC as an option.

Responses to the Comments of the Reviewer 2

The work from Langie and co-workers propose an integrated capture and conversion system within the theme of CCU, called Reaction Swing Absorption (RSA). In RSA, CO₂ is captured from flue gas (3.5% CO₂) in the form of bicarbonate with a 3 M Triethylamine solution and then converted in a MEA electrolyzer with the help of a BPM. The system is indeed very interesting and elegant and avoids the CO₂ compression and regeneration steps which limit the valorisation of eCO₂R systems. The description is detailed, clear and of interest of an overall scientific audience. The technoeconomic analysis (TEA) is an added asset that increases the quality of the manuscript, as it is often demanded as a baseline or validation of a technological strategy. Nevertheless, I still have a major concern about the work from Langie et al., which is the lack of discussion with other competing integrated strategies such as the direct bicarbonate electrolysis from DAC solutions (i.e., using KOH as absorbent). As I mention below in my comments, the state-of-the-art (which is outdated in this manuscript) presents conversion rates overall higher compared to this study. In addition, I have other minor comments that might improve the quality of the manuscript, listed below.

Overall, the work is of interest of Nature Communications and I believe the authors can give proper answers to my concerns in a revised manuscript. Nevertheless, due to my major concern I need to revise the manuscript a second time after revisions.

Major concern:

1. (Reviewer's Comment) *Line 48: The authors mention that bicarbonate electrolytes are proposed as direct-capture CO₂ conversion systems. They cite the work of Li et al. from Prof. Berlinguette's research group. Specifically, that their process showed 37% FE CO at 100 mA cm⁻². This reference is outdated. From 2019 to 2022, the same research group published two major papers on the same topic (bicarbonate electrolysis to CO/syngas), reporting, for instance:*

o 50% FE CO at 100 mA cm⁻² (ACS Energy Lett. 2020, 5, 7, 2165–2173)

o 59% (1 atm) and 95% (4 atm) FE CO at 100 mA cm⁻² (Energy Environ. Sci., 2022,15, 705-713).

This presents a huge increment in the energy efficiency of the bicarbonate electrolysis process and thus should be considered a competitor to the technology proposed in this study, where the max. FE CO at 100 mA cm⁻² is around 35% (using Coral-Ag/C). State-of-the-art references as the ones I mentioned before cannot be neglected in this study, even more when they present higher conversion rates. A TEA for bicarbonate electrolysis to be compared to the RSA is of course not needed, but the authors should clearly discuss why their approach is better than the proposed in the state-of-the-art.

(Authors' Response) Thank you for the valuable comment with very relevant papers. The bicarbonate electrolysis in the literature has been updated in the revised manuscript.

a) Reference

In the introduction section, we tried to describe the history of the study aimed at the direct utilization of bicarbonate. To our knowledge, since Li et al. paper from Prof. Berlinguette is the first research paper to implement the system for direct bicarbonate utilization, so we thought it would be meaningful to comment on the paper. As reviewer's comment, we revised the introduction section by emphasizing the meaning of the paper and added the additional recommended references from prof. Berlinguette group.

Changes made:

- Introduction section

Original: Li et al. shed light on the direct-capture CO₂ conversion system by suggesting bicarbonate as a valid option for converting captured CO₂; this process showed a 37% FE for CO at 100 mA cm⁻² without the addition of supporting electrolyte.

Revision: Li et al. shed light on the direct-capture CO₂ conversion system by being the first to suggest bicarbonate as a valid option for converting captured CO₂; this process showed a 37% FE for CO at -100 mA cm⁻² without the addition of supporting electrolyte, and the performance has further increased to 95% FE for CO at -100 mA cm⁻² under 4 atm pressure condition in a recent follow-up study.

b) Bicarbonate electrolysis

Reviewer recommended us to clearly present the advantages of the RSA system compared to the state-of-the-art references that directly utilize KHCO₃ as an electrolyte. Considering that KOH should be used to capture CO₂ into KHCO₃, this discussion eventually requires a comparison of KOH and TREA as a CO₂ absorbent.

KOH have been highlighted as a CO₂ absorbent in direct electrochemical conversion of captured CO₂ because KHCO₃ has been the most preferred electrolyte in electrochemical CO₂ reduction research field. KOH has an advantage of selectively absorbing CO₂ into KHCO₃ as shown in the following equations. However, KOH is barely applied in aqueous CO₂ capture processes due to the salt formation and corrosion issue (due to its extremely alkaline condition), which leads to the difficulty of continuous long-term operation. On the other hand, numerous amine-based CO₂ capture systems are industrially available.

Especially, the most state-of-the-art references utilizing KHCO₃ as electrolyte reported the detection of unreacted CO₂ along with the eCO₂R product by gas analysis [*Joule*, 2019, 3, 1487; *ChemElectroChem* 2021, 8, 2094; *Energy Environ. Sci.*, 2022, 15, 705]. However, in our RSA system, we didn't detect unreacted CO₂ from TREA solution during eCO₂R (already mentioned in Figure S3). This means that TREA could be a more favorable electrolyte for achieving high purity of syn-gas products, which could exclude product conditioning processes for CO₂ separation from syn-gas product. We believe this can be a key advantage of RSA system compared to the state-of-the-art bicarbonate electrolysis systems.

Other comments:

2. (Reviewer's Comment) *General comment: TEA is commonly used for triethanolamine instead of triethylamine. Triethanolamine is a common amine sorbent used in CO₂ capture research. The reaction between CO₂ and triethanolamine leads to carbamates, which is not related to the current study. I recommend the authors, to avoid confusions, use another acronym for triethylamine, such as TREA or TEYA.*

(Authors' Response) Thank you for the suggestion. In the revised manuscript, we used TREA for the abbreviation of trimethylamine. Instead, TEA was used for the abbreviation of techno-economic analysis.

3. (Reviewer's Comment) Flue gas is artificially prepared with CO_2 , N_2 and O_2 . However, in real flue gas other gases such as NO_x and SO_x , and solid particles are present as well. Although the amount is little, they might be significant for CCU approaches. What do the authors think about this? Have these traces an effect in the system? I understand that these traces are not added for the experiments, but the authors should mention this assumption in the Methods section.

(Authors' Response) Thank you for the valuable comment. This assumption is added in the revised manuscript. We are aware of the difference between real flue gas with impurities and experimentally controlled flue gas that can result in different conclusion. Actually, a paper (*Energy Environ. Sci.* 2022, 15, 705) shows a large impact of NO_3^- on FEs. However, our TREA electrolyzer is at its early development stage so we didn't consider impurities yet. We are planning to investigate it after improving the electrolyzer's performance.

Fig. 5 The impact of flue gas impurities on bicarbonate electrolysis performance. (a) FE_{CO} and FE_{H₂} measured during constant current density electrolysis for 3.0 M KHCO₃ solutions containing 100 ppm impurities. (b) Plot showing the decrease in FE_{CO} and FE_{H₂} after adding 100 and 500 ppm of NO₃⁻ to the 3.0 M KHCO₃ solution, but refreshing the electrolyte with pure 3.0 M KHCO₃ recovers the CO₂RR activity. These results show that the negative effect of NO₃⁻ on FE_{CO} is reversible. All electrochemical tests were performed at a constant applied current density of 100 mA cm⁻² with the porous metal electrode.

Changes made:

- The Subsection of Electrochemical measurement in the Methods section

Original: 3 M TEA (Sigma-Aldrich, 99%) was used for main CO₂ capture solution and 1 M KOH (Sigma Aldrich, >90%) was employed as anolyte. Before CO₂R, 3 M TEA solution was saturated by CO₂ for 1 h.

Revision: 3 M TREA (Sigma-Aldrich, 99%) was used as the CO₂ capture solution and 1 M KOH (Sigma Aldrich, >90%) was employed as anolyte. It is assumed that the TREA solution absorbs only CO₂ from the flue gas. Therefore, 3 M TREA solution was saturated with CO₂ for 1 h before CO₂R.

4. (Reviewer's Comment) *Proposing bicarbonate as the CO₂-supplier for eCO₂R has the advantage of avoiding amine during the capture step (and thus the formation of the undesirable C-N bond), since KOH is used as capture agent (KOH+CO₂ = KHCO₃). In fact, this method is proposed not only for flue gas capture but for Direct Air Capture, too (much lower %CO₂, 0.04%). It is true that, up to day, there are no reports of electrolysis of real absorbed KHCO₃ solutions but instead pre-fabricated KHCO₃ solutions were used as electrolyte for bicarbonate electrolysis proof-of-concept experiments (See ref. 14 from manuscript or ChemElectroChem, 9(5), e202101540 (2022)). The authors mention the necessity of using amine sorbents in addition to KOH or H₂O for generating bicarbonate from CO₂ during the capture step. What is the reasoning of this? What is the real advantage of using TEA? If I understood properly, it matters with the mechanism shown in Figure 2. But then, aren't other amines (that don't produce carbamates) able to fulfill this job? The authors should discuss the choice of amine over other potential absorbents and the use of TEA as bicarbonate formation promotor.*

(Authors' Response) Thank you for the comment.

We think bicarbonate is essential for direct CO₂ conversion, because carbamate is not appropriate CO₂ source for this. As Reviewer 2 mentioned, KOH can provide bicarbonate but the papers (ref. 14 in the manuscript and ChemElectroChem, 2022, 9, e202101540) suggest thermochemical purification of CO₂ before entering electrolyzers. Although KOH can be used for direct CO₂ conversion, it has been reported that metal hydroxides can cause numerous problems when used as an eCO₂R electrolytes. Especially, the salt precipitation, bicarbonate deposition on electrodes, and corrosion issue of KOH electrolyte often cited for common obstacles for the long-term operation of eCO₂R system. As a result, we propose a practical method to overcome drawbacks of the metal hydroxide electrolytes. Commonly used tertiary alkanol amine (e.g., triethanol amine) can also generate the bicarbonate but sudden viscosity increase on these amine hinders mass transfer of CO₂ to the catalyst surface.

In this study, we demonstrate that TREA is not only promising electrolyte for eCO₂R but also has low concentration CO₂ absorption capability by performing pilot scale column experiment. Now we are thinking of TREA for Direct Air Capture system.

5. (Reviewer's Comment) Line 64: The authors claim that in RSA the release of syngas from the eCO₂R is due to the low solubility of the product in the amine solvent. Here, I have two comments:

o I agree about the thermodynamic reason (solubility), however, what about the kinetics? Isn't the release of product a limiting step? How did the authors study this? I.e., the solubility might be low, and the release of product thermodynamically favored but on the other hand the kinetics slow and therefore the eCO₂R is limited not anymore by the electrochemical reaction but in fact by the release of product.

o Shouldn't this statement be the same for other eCO₂R-to-syngas systems using carbamate as CO₂ source and amine solvents as electrolytes? Why is in RSA the low solubility of syngas a specific advantage compared to conventional carbamate electrolyzers?

(Authors' Response) Thank you for the careful comments. As reviewer mentioned, since CO and H₂ have very low solubility, we expected that syn-gas would not cause the problem regarding low release kinetics from aqueous amine solution. This expectation is valid not only in our RSA system (using triethylamine), but also in the previously studied carbamate electrolyzer (using monoethanolamine) based on the data from previous studies. In previous carbamate electrolyzer studies using 30 wt.% monoethanolamine solvent (*ChemSusChem*, 2017, 10, 4109; *Nature Energy*, 2021, 6, 46), the sum of Faradaic efficiencies of products including CO and H₂ under various experimental conditions were reported to be almost 100%. In our RSA system, total Faradaic efficiency of CO and H₂ was constantly calculated nearly 100%, even during 70 h of long-term operation (Figure). This indicated that the release kinetics of CO and H₂ from aqueous amine solvents were rapid enough. Consequently, both RSA system and carbamate electrolyzers have low solubility of CO and H₂, which is not the particular advantage of RSA system. Nevertheless, the limitation of carbamate electrolyzer still remains as mentioned in the manuscript; since the carbamate has a strong C-N bond, it is too difficult to achieve reactant CO₂ from carbamate during eCO₂R without additional additives such as alkali metal cations (LiCl, NaCl, KCl, etc.) and surfactants (CTAB).

Figure. long-term eCO₂R performance represented as CO FEs (red), H₂ FEs (black) and cell voltages (blue) for coral-Ag/C at -100 mA cm⁻² in 3 M TREA.

6. **(Reviewer's Comment)** Line 100: C-NMR showed a single peak derived from bicarbonate and carbonate. However, carbonate is not mentioned anymore in the manuscript and instead all the carbon present in solution is considered bicarbonate. How were the authors sure that carbonate is not present in solution? The concentration of carbonate is relevant for the electrolysis step, as 2 protons must be released from the BPM to convert carbonate to CO₂ (while only one is needed in the case of bicarbonate).

(Authors' Response) In ¹³C-NMR, since it is difficult to distinguish the bicarbonate and carbonate peaks, we notated the single peak as bicarbonate and/or carbonate ions. Nevertheless, as following the CO₂ capturing mechanism in TREA aqueous solution, the bicarbonate would be the main CO₂ captured form. Furthermore, considering that pH of CO₂ saturated 3 M TREA solution is 7.68 during eCO₂R, the major species presented in 3 M TREA is bicarbonate based on following chemical reactions.

Where, pH is 7.68, and pKa₂ is 10.25 (at 25°C). Based on the calculation, the concentration of [HCO₃⁻] is about 371 times higher than that of [CO₃²⁻]. According to our analysis, since HCO₃⁻ concentration in 3 M TREA is 2.78 M, the concentration of CO₃²⁻ is 7.48 mM, which is expected to have a negligible influence as a reactant. To avoid confusion for the readership, we revised the manuscript as below.

Changes made:

- On page 4, line 102

Original: In the ¹³C-NMR spectrum of CO₂ captured in a 3 M TEA/H₂O solution, a single peak derived from bicarbonate and carbonate ions was observed. Carbamate species were not observed, which indicates that bicarbonate/carbonate pairs were selectively generated during absorption of CO₂ by TEA.³²

Revision: In the ¹³C-NMR spectrum of CO₂ captured in a 3 M TREA/H₂O solution, a single peak derived from bicarbonate (and/or carbonate) ion was observed. Carbamate species were not observed. Considering that the CO₂ saturated 3 M TREA/H₂O is a neutral solution (pH 7–8), it can be deduced that bicarbonate was selectively generated during absorption of CO₂ by TREA according to the chemical equilibrium determined by pH.³⁷

Reviewer Comments, second round -

Reviewer #1 (Remarks to the Author):

I thank the authors for considering my comments and revising the manuscript accordingly. Most of my comments have been addressed sufficiently, but I still have some (minor) comments before the manuscript can be accepted for publication.

These comments are:

1. My original comment 3: I agree with the authors that some KOH is lost due to (bi)carbonate formation. The authors further mention that: KOH is considered to be a currently available CO₂ absorbent, but KOH is barely applied in aqueous CO₂ capture processes... I would like to note that one of the leading direct air capture companies (Carbon Engineering) is using KOH as a solvent for CO₂ capture from air.

2. My original comment 4: The authors conclude that "the developed CO₂ utilization pathways successfully mitigate GWPs, none of the proposed CCU systems are economically competitive with conventional production pathways'. The authors base this conclusion on a range of selected papers mainly on methanol production. As shown by Centi et al., Economics of CO₂ utilization: A critical analysis, Front. Energy Res., 2020, 8, 1-16, methanol production from CO₂ may be economical or not depending on which source is selected and which assumptions have been used. In my opinion, the discussion of methanol is irrelevant and could be excluded from the paper, since the paper is mainly concerned about syngas production. If the authors decide to retain the cumbersome methanol comparison, then they should carefully read the paper of Centi et al.

3. My original comment 10: The authors replaced figure 3g with a new figure evaluated at a current density of 100 mA/cm² instead of 150 mA/cm². This resulted in a cell voltage of 3.5 V instead of 4.5 V and the H₂:CO ratio was maintained to 2:1. However, the FE of CO formation is still relatively low 35%, which means that mainly hydrogen is produced. The authors provide a cost comparison for hydrogen production from their CO₂ electrolyzer and state-of-the-art hydrogen (PEM) electrolyzer. I disagree with the conclusions of the authors that "for both current and optimistic scenarios, the CO₂ electrolyzer is a better option for H₂ supply". This simply cannot be true because commercial hydrogen electrolyzers operate around 2V and much higher current densities than the CO₂ electrolyzer of the authors. Therefore, the cost of hydrogen per unit of mass will be lower for the commercial electrolyzer, since the required electrolyzer area (CAPEX) and the required power (OPEX) will be lower. Moreover, the capital cost of the CO₂ electrolyzer will be much higher than that of a water electrolyzer due to increased complexity. The authors seem to neglect the capital cost and include an additional cost for water electrolysis, which makes the CO₂ electrolyzer more favorable. The authors mention that the additional cost includes compression, storage, and dispensing. All these costs are not present for an on-site hydrogen electrolyzer, which can be used to adjust the H₂:CO ratio. PEM electrolyzers can operate at high pressures to serve high pressure downstream processes. Costs of compression, storage, and dispensing are mainly relevant for refueling stations. The authors should provide a cost comparison on a same basis.

4. My original comment 13: I disagree with the authors that the SOEC based CCU technologies are not economically feasible and have a short-term CCU application. In fact, the Haldor-Topsoe process and the Sunfire co-electrolysis process are (near) commercial (they have much higher TRLs than the RWGS and the low temperature processes considered by the authors). Furthermore, I disagree with the authors that a large amount of heat required in the SOEC emits a lot of CO₂. These SOECs operate close to the thermoneutral voltage, thus the heat demand for the reaction itself is not so high. Pre-heating can be achieved by e.g., burning hydrogen instead of methane without additional CO₂ emissions. Proper heat integration can make SOECs very efficient for CCU applications now and in the future.

The paper may be accepted after the authors carefully revise the manuscript.

Reviewer #2 (Remarks to the Author):

Review of the Revised manuscript "Toward economical application of carbon capture and utilization technology with near zero carbon emission" of Lengje et al.

In the revised manuscript the authors did a dedicated exercise to answer the remarks of the reviewers. The literature has been updated and some of the assumptions that were not initially clear have been clarified. The quality of the manuscript has thus substantially increased. I think the manuscript is ready to be published after the authors cover a small concerns I still have.

Concerning comment 3.

Although your TREA electrolyzer is at its early development stage that doesn't exclude that you can add a brief discussion about the impact of the impurities on FEs. In fact, in the response letter you mention that NO_3^- shows large impact on FE. The changes made to the experimental section are not enough and the reader might reach the conclusion that these impurities are not being considered for upscaling steps. You could include the very same explanation and reference you provided to us to the main manuscript and, if you are considering to investigate these effects in future steps, mention in as future perspectives.

Responses to the Comments of the Reviewer 1

I thank the authors for considering my comments and revising the manuscript accordingly. Most of my comments have been addressed sufficiently, but I still have some (minor) comments before the manuscript can be accepted for publication. The paper may be accepted after the authors carefully revise the manuscript.

These comments are:

1. (Reviewer's Comment) *My original comment 3: I agree with the authors that some KOH is lost due to (bi)carbonate formation. The authors further mention that: KOH is considered to be a currently available CO₂ absorbent, but KOH is barely applied in aqueous CO₂ capture process... I would like to note that one of the leading direct air capture companies (Carbon Engineering) is using KOH as a solvent for CO₂ capture from air.*

(Authors' Response) Thank you for your detail comment and we agree that KOH can be a promising solution for low concentration CO₂ capture process. Per your comment, we modified our original sentence.

Changes made:

Original Sentence: In this system, KOH is considered to be a currently available CO₂ absorbent, but KOH is barely applied in aqueous CO₂ capture processes due to the salt formation and corrosion issues caused by the extremely alkaline condition.¹⁷⁻²⁰

Revision: In this system, KOH is considered to be a currently available CO₂ absorbent. Although KOH is a promising solvent for CO₂ capture and conversion, the salt formation and corrosion issues caused by the extremely alkaline condition are considered as major challenges for the commercialization.¹⁷⁻²⁰

2. (Reviewer's Comment) *My original comment 4: The authors conclude that “the developed CO₂ utilization pathways successfully mitigate GWPs, none of the proposed CCU systems are economically competitive with conventional production pathways”. The authors base this conclusion on a range of selected papers mainly on methanol production. As shown by Centi et al., Economics of CO₂ utilization: A critical analysis, Front. Energy Res., 2020, 8, 1–16, methanol production from CO₂ may be economical or not depending on which source is selected and which assumptions have been used. In my opinion, the discussion of methanol is irrelevant and could be excluded from the paper, since the paper is mainly concerned about syngas production. If the authors decide to retain the cumbersome methanol comparison, then they should carefully read the paper of Centi et al.*

(Authors' Response) Thank you for your valuable comment. As per your comment, we decided to remove Fig. 1(b) from the main manuscript, and relocated it to the Supplementary Information. The original text regarding Fig. 1(b) was relocated along with the figure, and additional information was provided to take in the contents of Centi et al.

Changes made in pages S29 and S32 of Supplementary Information,
(additional text, added during the relocation process from the main manuscript to the Supplementary Information, is shown in red.)

In Fig. S21, the RCI and GWP of the different studies are evaluated based on the following criteria: first, the $C_{\text{Product}_{\text{CCU}}}$ value is determined as the base case cost value provided by the specific literature. Most studies provide cost values for various scenarios, such as changes in renewable electricity costs, and policy changes such as carbon credits. These changes can greatly alter the $C_{\text{Product}_{\text{CCU}}}$ values, but inherit large uncertainties in terms of the implementation period, or the level of implementation. To allow a fair comparison, the base case of the proposed process/system, which is based on a realistic scenario, is taken as the $C_{\text{Product}_{\text{CCU}}}$ value of that study. Secondly, the GWP value is calculated differently according to the range of the system boundary designated within the study. When a full system boundary is implemented, where the CO₂ source is included along with the CCU process, the GWP values are calculated by subtracting the conventional GWP value from the GWP of the proposed process. For a restricted system boundary, where only the proposed process is selected as the system of analysis, neglecting the CO₂ source, the GWP value provided within the study is used. Thirdly, all of the GWP values provided within the references are regarded as GWP100 values, which indicates the GWP within a 100-year period. While some studies state the use of GWP100 during LCA, most of the studies do not specify the LCA evaluation method of use or the GWP evaluation period. Since the most widely used LCA evaluation method is the ReCiPe 2016 Hierarchy method, which evaluates the GWP100 of various chemicals, it is assumed that all of the studies are evaluated based on this method. **It should be noted that the considered list of references is non-exhaustive, and certain studies conducting techno-economic analysis can show profitability according to the different assumptions made.**

Figure S21. RCI and GWP for syngas-based chemical production from CCU pathways in the literature. While CO₂ is mitigated in most CCU pathways, none of the pathways are economically competitive compared to conventional chemical production.

3. (Reviewer’s Comment) My original comment 10: The authors replaced figure 3g with a new figure evaluated at a current density of 100 mA/cm² instead of 150 mA/cm². This resulted in a cell voltage of 3.5V instead of 4.5V and the H₂:CO ratio was maintained to 2:1. However, the FE of CO formation is still relatively low 35%, which means that mainly hydrogen is produced. The authors provide a cost comparison for hydrogen production from their CO₂ electrolyzer and state-of-the-art hydrogen (PEM) electrolyzer. I disagree with the conclusions of the authors that “for both current and optimistic scenarios, the CO₂ electrolyzer is a better option for H₂ supply”. This simply cannot be true because commercial hydrogen electrolyzers operate around 2V and much higher current densities than the CO₂ electrolyzer of the authors. Therefore, the cost of hydrogen per unit of mass will be lower for the commercial electrolyzer, since the required electrolyzer area (CAPEX) and the required power (OPEX) will be lower. Moreover, the capital cost of the CO₂ electrolyzer will be much higher than that of a water electrolyzer due to increased complexity. The authors seem to neglect the capital cost and include an additional cost for water electrolysis, which makes the CO₂ electrolyzer more favorable. The authors mention that the additional cost includes compression, storage, and dispensing. All these costs are not present for an on-site hydrogen electrolyzer, which can be used to adjust the H₂:CO ratio. PEM electrolyzers can operate at high pressure to serve high pressure downstream processes. Costs of compression, storage, and dispensing are mainly relevant for refueling stations. The authors should provide a cost comparison on a same basis.

(Authors’ Response) Thank you for the response and valuable comment. The authors have conducted totally new TEA, LCA, and GSA studies, based on this comment. As a result, all figures, tables, and contexts of modeling part in the secondly revised manuscript have been totally rewritten.

a) Alternative H₂ production

An on-site H₂ production by a PEM electrolyzer replaces the external H₂ purchase, so there is no compression, storage, and dispensing (CSD) cost. In spite of no CSD cost, there still remain some additional costs. Production cost without CSD cost in the figure below (Fig. R1) consists of the electricity cost (blue bar) and others for CAPEX, fixed operating cost, etc.

Figure R1. H₂ production cost by PEM electrolyzer (James et al. DoE, 2016).

The reference of Fig. R1 provides the amount of electricity usage and additional cost (see the table below, Table R1). Based on this, it can be postulated that H₂ from PEM is equivalent to produce at 2.5–4.0V due to the additional cost. For example, the additional cost of \$1.743/kg is same with 25.5 and 39.2 kWh/kg for the energy mix (\$0.068/kWh) and solar (\$0.045/kWh) cases in the current scenario, respectively. These amounts of electricity are equivalent to additional voltage of 0.959 and 1.472V, respectively.

Table R1. Equivalent cell voltage of PEM electrolyzer considering additional cost.

	Current			Optimistic		
	Energy Mix	Solar	Wind	Energy Mix	Solar	Wind
Electricity (kWh/kg)	54.3			50.2		
Voltage (V)	2.0422			1.888		
Elec Price (\$/kWh)	0.068	0.045	0.044	0.043	0.018	0.013
Elec Cost (\$/kg)	3.712	2.417	2.389	2.164	0.890	0.658
Additional (\$/kg)	1.743			0.741		
Additional Volt (V)	0.959	1.472	1.489	0.647	1.573	2.127
Equivalent Volt (V)	3.001	3.514	3.532	2.535	3.461	4.015

b) Optimal CO Faradaic efficiency

In the H₂ production point of view, high CO FE in the bicarbonate electrolyzer seems economically feasible, because H₂ can be produced by the more efficient (lower cell voltage) PEM electrolyzer than the bicarbonate electrolyzer. However, the determination of optimal CO FE is quite complex. Some factors should be considered:

- Alternative H₂ production by PEM needs some costs in addition to electricity usage, leading to higher equivalent cell voltage of 2.5–4.0V.
- To increase amount of H₂ from PEM, the bicarbonate should be operated at lower current density in order for higher CO FE. This makes lower electricity usage, higher cell area (larger CAPEX), and lower production cost of byproduct H₂ in the bicarbonate electrolyzer.

Figure R2. Performance of the bicarbonate electrolyzer in current and optimistic scenario.

The actual cell voltage of the optimized bicarbonate electrolyzer is similar to that of equivalent voltage of PEM. At this voltage, there is no discrepancy between two electrolyzers in terms of H₂ production cost. In the current scenario, CO FE at this voltage is quite low. However, it is uneconomical to increase the CO FE by lowering the cell voltage due to already extremely large CAPEX with very low current density (42–72 mA/cm²). As the bicarbonate electrolysis technology is improving up to the gas CO₂ electrolyzer level (optimistic scenario), the optimal CO FE becomes 90% (upper bound in this study) with much lowered CAPEX in the secondly revised manuscript.

Table R2. Comparison of cell voltage of PEM and optimized bicarbonate electrolyzer.

	Current			Optimistic		
	Energy Mix	Solar	Wind	Energy Mix	Solar	Wind
Equivalent Volt (V)	3.001	3.514	3.532	2.535	3.461	4.015
Bicarbonate Volt (V)	3.11	3.44	3.43	2.74	3.1	3.1
CO FE	0.51	0.41	0.41	0.9	0.9	0.9

Figure S18. Capital investment of three processes under various energy sources. (a) Energy mix case.

Changes made:

Original Sentence (Main Manuscript): We carried out a comprehensive techno-economic analysis (TEA) and an LCA of RSA using a process model based on the abovementioned experimental results. The result was compared with those of conventional processes using the RWGS and gas phase eCO₂R (Fig. 1) to confirm the superiority of the proposed RSA process. A global sensitivity analysis (GSA) was also performed to assign cost contributions and GWPs and eventually to highlight the crucial factors for further improvement of RSA.⁵¹ Current and optimistic scenarios were compared through analysis of several related factors: *i*) the electricity generation cost based on different energy sources, *ii*) improvements to the bicarbonate electrolyzer, *iii*) purchase cost of H₂ produced from water electrolysis as an alternative option to satisfy the H₂-to-CO ratio of 2 for syngas, and *iv*) CO₂ capture rate of the absorber. H₂ in

syngas comes from purchase, from the CO₂ electrolyzer as a by-product, or from both. There is competition between internal production and external purchase. Therefore, the third factor (purchase cost of H₂) is added for the sensitivity analysis. Whereas the operating conditions of RWGS and gas eCO₂R processes are well known, those of the RSA should be optimized due to its early development stage. As a result, the optimal cell voltage, which is pertinent to the CO FE and current density, was found to minimize the break-even price of syngas. The details of process modeling, TEA, and LCA are explained in the Supplementary Note 4.

As illustrated in Fig. 4a, 4b, S17, and S18, the RSA process outperforms the other two CCU processes, in terms of operating expenditures and the break-even price of syngas in both current and optimistic scenarios, regardless of energy sources. Because the current bicarbonate electrolyzer in the RSA is inefficient, it is operated at low current density. Therefore, the bicarbonate electrolyzer cost is similar to the total cost of the CO₂ electrolyzer and the unreacted CO₂ separation system for the gas eCO₂R process. As the bicarbonate electrolyzer is improved in the optimistic scenario, its cost can be reduced considerably. Unlike the RWGS and gas eCO₂R processes that purchase H₂ and have high portion of the material cost in operating expenditure (OPEX), the RSA process produces H₂ by the bicarbonate electrolyzer so it shows high portion of the utility cost in OPEX. When renewable energy is used, the optimistic break-even prices for syngas will be dropped to \$0.58/kg of syngas for wind and \$0.47/kg of syngas for solar. These prices can compete with fossil fuel-based syngas processes (Fig. 4f). Although the wind case has a slightly higher syngas price than that in the solar case, wind is a more promising energy source from an environmental perspective (only a quarter of the GWP100 value in comparison to the solar case). Fig. 4c presents the system boundaries for LCAs containing CO₂ sources within the boundaries. Two important impacts for an LCA of CCU technology are GWP100 and fossil resource scarcity (FRS). In the case of GWP100, the RSA process has a lower impact than the other processes in most cases. The GWP100 of the RSA can be minimized to 0.30 kg CO₂ eq./kg syngas (Fig. 4d) when wind energy is used for electricity generation. However, GWP100 is highly sensitive to the energy source, so this value can be increased up to 6.60 kg CO₂ eq./kg syngas in the energy mix case (Fig. 4e). The FRS result shows a similar trend to GWP100, because the RSA is highly energy intensive process.

A GSA was also conducted to support decision-making for establishing appropriate CCU strategies and policies and providing the priority of research targets. Unlike local sensitivity analyses, such as a one-factor-at-a-time method, GSA changes all uncertain input variables simultaneously and monitors the variances of dependent variables.⁵² Thus, a GSA generates results in the form of Sobol indices, which indicate the impacts of uncertain variables on dependent variables. There are six uncertain variables in the RSA for the solar and wind cases: CO₂ capture rate, CO FE improvement, current density improvement, unit cost of the electrolyzer, unit cost of electricity generation, and water electrolysis improvement for external H₂ production. For the energy mix case, portions of solar and wind energies are added to the uncertain variable set.

Fig. 4g compares the 1st order Sobol indices of the break-even price of syngas and the GWP100 for the RSA process under a given range of uncertain variables from current to optimistic scenarios. When wind energy is used for the RSA, the most sensitive economic

factor is the electricity generation price (energy generation improvement in Fig. 4g). Wind energy is well-known for its very low CO₂ emission, so electricity becomes no longer a major environmental issue. Consequently, uncaptured CO₂ to be released to air in the absorber is the main contributor to CO₂ emission when wind energy is used. This result indicates that the near net-zero CO₂ emission is achievable as the CO₂ capture rate increases in the chemisorption process. For the energy mix case, no factor dominantly impacts the break-even price. From an environmental perspective, the additional cost related to external H₂ production, such as H₂ compression, storage, and dispensing, are the most significant factors. In other words, the origin of H₂ (internal production or external purchase) in syngas determines electricity usage that dominates CO₂ emission in the energy mix case. The detail results for different scenarios, cases, and dependent variables can be seen in the Supplementary Note 4.

Revision (Main Manuscript): We carried out a comprehensive techno-economic analysis (TEA) and an LCA of RSA using a process model based on the abovementioned experimental results. The result was compared with those of conventional processes using the RWGS and gas phase eCO₂R (Fig. 1) to confirm the superiority of the proposed RSA process. A global sensitivity analysis (GSA) was also performed to assign cost contributions and GWPs and eventually to highlight the crucial factors for further improvement of RSA.⁵¹ Current and optimistic scenarios were compared through analysis of several related factors: *i*) the electricity generation cost based on different energy sources, *ii*) improvements to the bicarbonate electrolyzer, *iii*) purchase cost of H₂ produced from an on-site water electrolysis system as an alternative option to satisfy the H₂-to-CO ratio of 2 for syngas, and *iv*) CO₂ capture rate of the absorber. H₂ in syngas comes from purchase, from the CO₂ electrolyzer as a by-product, or from both. There is competition of H₂ supply between the bicarbonate electrolyzer and on-site purchase. Therefore, the third factor (purchase cost of H₂) is added for the sensitivity analysis. Whereas the operating conditions of RWGS and gas eCO₂R processes are well known, those of the RSA should be optimized due to its early development stage. As a result, the optimal cell voltage, which is pertinent to the CO FE and current density, was found to minimize the break-even price of syngas. The details of process modeling methodology and TEA and LCA results are explained in the Supplementary Note 4.

As illustrated in Fig. 4a, 4b, S18, and S19, the RSA process outperforms the other two CCU processes, in terms of operating expenditure (OPEX) and the break-even price of syngas in the optimistic scenario, regardless of energy sources. In the current scenario, all three processes have the similar OPEX and break-even price, although the RSA shows an enormous capital expenditure (CAPEX) due to the inefficient bicarbonate electrolyzer. However, current density can be increased at lowered voltage with higher CO FE, as the performance of bicarbonate electrolysis is improved in the optimistic scenario. This improvement considerably reduces CAPEX of the RSA and converts the H₂ supply from the bicarbonate to water electrolyzer. Considering lower cell voltage in water electrolysis than that in bicarbonate electrolysis, the H₂ supply change results in the RSA beating other two processes for CAPEX, OPEX, and break-even price in the optimistic scenario. When renewable energy is used, the optimistic break-even prices for syngas will be dropped to \$0.65/kg of syngas for wind and \$0.56/kg of syngas for solar. These prices can compete with fossil fuel-based syngas processes

(Fig. 4f). Although the wind case has a slightly higher syngas price than that in the solar case, wind is a more promising energy source from an environmental perspective (only **one-third** of the GWP100 value in comparison to the solar case).

Fig. 4c presents the system boundaries for LCAs containing CO₂ sources within the boundaries. Two important impacts for an LCA of CCU technology are GWP100 and fossil resource scarcity (FRS). In the case of GWP100, the RSA process has a lower impact than the other processes in **all cases except the current energy mix case**. The GWP100 of the RSA can be minimized to **0.27 kg CO₂ eq./kg syngas** (Fig. 4d) when wind energy is used for electricity generation. However, GWP100 is highly sensitive to the energy source, so this value can be increased up to **5.52 kg CO₂ eq./kg syngas** in the energy mix case (Fig. 4e). The FRS result shows a similar trend to GWP100, because the RSA is highly energy intensive process.

A GSA was also conducted to support decision-making for establishing appropriate CCU strategies and policies and providing the priority of research targets. Unlike local sensitivity analyses, such as a one-factor-at-a-time method, GSA changes all uncertain input variables simultaneously and monitors the variances of dependent variables.⁵² Thus, a GSA generates results in the form of Sobol indices, which indicate the impacts of uncertain variables on dependent variables. There are six uncertain variables in the RSA for the solar and wind cases: CO₂ capture rate, CO FE improvement, current density improvement, unit cost of the electrolyzer, **electricity generation improvement, and additional cost (except electricity cost) when H₂ in syngas is supplied by an on-site water electrolysis system, instead of H₂ production in the bicarbonate electrolyzer as by-product**. For the energy mix case, portions of solar and wind energies **in electricity generation mix** are added to the uncertain variable set.

Fig. 4g compares the 1st order Sobol indices of the break-even price of syngas and the GWP100 for the RSA process under a given range of uncertain variables from current to optimistic scenarios. When wind energy is used for the RSA, the most sensitive economic factor is the electricity generation price (energy generation improvement in Fig. 4g). Wind energy is well-known for its very low CO₂ emission, so electricity becomes no longer a major environmental issue. Consequently, uncaptured CO₂ to be released to air in the absorber is the main contributor to CO₂ emission when wind energy is used. This result indicates that the near net-zero CO₂ emission is achievable as the CO₂ capture rate increases in the chemisorption process. For the energy mix case, no factor dominantly impacts the break-even price. From an environmental perspective, the additional cost related to **the on-site water electrolysis system is the most significant factor**. In other words, the improvement of water electrolysis infrastructure and technology economically attracts this system for H₂ production and substantially reduces the electricity usage. Consequently, the amount of emitted CO₂ in the energy mix case can be largely decreased by this improvement. The detail TEA, LCA, and GSA results for different scenarios and cases can be seen in the Supplementary Note 4.

Original Sentence (Figures in Main Manuscript):

Figure 4. Results of techno-economic analysis, LCA, and GSA. (a) Capital expenditures for three processes. (b) Operating expenditures and break-even prices for syngas from the three processes. (c) Graphical representation of LCA system boundaries. (d) LCA results for the three processes when using wind energy. The left and right bars in each scenario represent global warming potential (GWP) and fossil resource scarcity (FRS), respectively. (e) LCA results for the RSA process with different cases and scenarios. (f) Cash flow charts for wind electricity with an optimistic scenario. (g) 1st order Sobol indices from global sensitivity analysis.

Figure 5. Syngas as a versatile intermediate chemical having a competitive price when using the eco-friendly CO₂ RSA process. Estimated levelized cost of target chemicals produced from consecutive RSA and downstream processes⁵³ (red), current market prices of chemicals (blue), and heat of reactions of syngas production and downstream reactions^{54,55} (green). (Market prices from <https://www.methanex.com/our-business/pricing>, <https://tradingeconomics.com/commodities>, https://repository.upenn.edu/cgi/viewcontent.cgi?article=1119&context=cbe_sdr, and <https://www.globalpetrolprices.com/>)

Revision (Figures in Main Manuscript):

Figure 4. Results of techno-economic analysis, LCA, and GSA for the wind case. (a) Capital expenditures for three processes. (b) Operating expenditures and break-even prices for syngas from the three processes. (c) Graphical representation of LCA system boundaries. (d) LCA results for the three processes. The left and right bars in each scenario represent global warming potential (GWP) and fossil resource scarcity (FRS), respectively. (e) LCA results for the RSA process with different cases and scenarios. (f) Cash flow charts for an optimistic scenario with a selling price of \$0.8/kg syngas. (g) 1st order Sobol indices from global sensitivity analysis.

Figure 5. Syngas as a versatile intermediate chemical having a competitive price when using the eco-friendly CO₂ RSA process. Estimated leveled cost of target chemicals produced from consecutive RSA and downstream processes⁵³ (red), current market prices of chemicals (blue), and heat of reactions of syngas production and downstream reactions^{54,55} (green). (Market prices from <https://www.methanex.com/our-business/pricing>, <https://tradingeconomics.com/commodities>, https://repository.upenn.edu/cgi/viewcontent.cgi?article=1119&context=cbe_sdr, and <https://www.globalpetrolprices.com/>)

Original Sentence (Supporting Information): The purpose of modeling in this study is to find an economically and environmentally feasible CCU technology in order for the replacement of thermochemical technology of CO₂ conversion. The three CCU processes considered in this study are thermal CO₂ conversion (RWGS, reverse water gas shift reaction), electrochemical CO₂ conversion (gas eCO₂R), and electrochemical bicarbonate conversion (RSA, reaction swing absorption) for syngas production. Solid oxide electrolysis cell (SOEC) can also be an option for syngas production [1], but SOEC requires a large amount of heat to be operated at high temperature (600–850°C) that causes a lot of CO₂ emission. Also, the major cell component for electrolyzer in SOEC system is an expensive ceramic material, such as yttria-stabilized zirconia, leading to large capital investment. Consequently, this did not consider SOEC as an option.

Flowsheets of these processes are illustrated in Fig. S13. Both the RWGS and gas eCO₂R processes require not only a stripping column for regeneration of amine but also pressure swing adsorption (PSA) system for separation of unreacted CO₂ and products (CO and H₂). Also, the reactor and PSA system are operated at different temperature in both the RWGS and gas eCO₂R processes so heat integration by heat exchanger network is needed. On the other hand, the RSA process uses neither the regeneration column nor PSA system, because unreacted CO₂ is still in the amine solution while low soluble products are in gaseous phase. Moreover, the RSA process has no need for heat integration, as it is operated under ambient condition. The RWGS

process produces CO only. To confirm H₂/CO ratio of 2 (desirable syngas for methanol) for product, the RWGS process is assumed to purchase H₂. The other two electrolysis-based processes can obtain H₂ from the electrolyzer as a by-product and/or from external purchase of H₂ produced by water electrolysis.

The process models are based on the literature as well as our experiment. The RWGS process is modeled, based on the result of the stable 80-hour operation of RWGS reaction from Sun et al. [2]. Cell voltage, current density and Faradaic efficiency (FE) of the gas eCO₂R are obtained from the experimental study of Liu, et al. [3] for over 3,000 hours. The PSA system for the RWGS and gas eCO₂R conversion processes is operated at 300°C [4] and its energy consumption, capital investment and operating cost are calculated by parameters from Jouny et al. [5]. When H₂ is externally purchased from outside of the process system, water electrolysis system is assumed to produce H₂ [6]. The experiment data of bicarbonate electrolysis and chemisorption by trimethylamine (TREA) are used for the RSA process. The relations among cell voltage, FE and current density in the bicarbonate electrolyzer are shown in Fig. S14.

...

LCAs with impact parameter values of materials, energies, etc. obtained from SimaPro V9.3 are conducted by Matlab, based on the process simulation results from Aspen Plus. The ReCiPe 2016(H) is used as a ‘Cradle-to-Gate’ method with the ecoinvent database V3.8 for the system boundaries in Fig. 4(c). For simplicity, transportation of raw materials (external H₂ and amines) is ignored for LCA. Because the three processes in this study have early technology maturity with different technology readiness levels, they are highly uncertain and lack of data availability [9–10]. Many LCA studies of CCU technology in the literature have been conducted for comparative assessment using ‘Cradle-to-Gate’ approach with functional unit of mass [11]. As a result, this study employed same methodology and conducted sensitivity analysis and scenario analysis.

...

As energy system has been changing from fossil to renewable and the bicarbonate electrolysis is currently at its early development, two scenarios are considered in this study: current and optimistic. In the optimistic scenario, i) the bicarbonate electrolyzer is improved, as seen in Fig. S14, ii) water electrolysis system for purchasing H₂ is advanced [6] and iii) electricity generation cost is reduced [13]. Also, there are three cases in each scenario, based on energy sources for electricity generation: energy mix, solar, and wind. Thus, total 18 results (2 scenarios * 3cases * 3 processes) can be obtained.

Table S4 lists parameters values in the current and optimistic scenarios. The electrolyzer-related parameters are FE improvement and CD improvement for bicarbonate electrolysis and unit cost of bicarbonate electrolyzers. The FE and CD improvements mean cell voltage reduction from current scenario at the same of FE and CD values (Fig. S14). The unit cost of electrolyzer in the current scenario comes from the study of Jouny et al. [5].

...

The external H₂ production by water electrolysis is based on the report of James et al. [6]. Its cost consists of electricity cost for electrolysis and additional cost for H₂ compression, storage and distribution. The report provides the amount of electricity consumption (54.3 vs. 50.2 kWh/kg H₂) and additional cost (\$4.18 vs. \$2.31/kg H₂) for current and future.

...

The operating expenditure (OPEX) for the three processes are shown in Fig. S19. The RWGS and gas eCO₂R processes purchase external H₂ so most of OPEX is material cost. In contrary, the optimal cell voltage in the RSA is high enough for low CO FE of 33–39% and high H₂ FE of 67–61%. Therefore, almost all H₂ comes from the bicarbonate electrolyzer, and the utility cost becomes the major component of OPEX. As the unit cost of electricity is cheaper in the optimistic scenario by different energy mix and lowered cost of renewable electricity, both OPEX and break-even price are decreased for all the processes. In the RSA process, additional factor for the break-even price reduction is the improvement of the bicarbonate electrolyzer (higher CD, lower the unit cost of equipment, etc.). When the RSA process uses renewable energy only for electricity, syngas price in the optimistic scenario are \$0.47/kg syngas for solar and \$0.58/kg syngas for wind, respectively, that can economically compete with the fossil-fuel-based conventional technology.

...

As a result, the RSA process with the wind case in the optimistic scenario seems the best selection of syngas production from the economic and environment points of view: \$0.58/kg syngas with CO₂ emission of 0.30kg CO₂ eq./kg syngas. When solar energy is used, the break-even price can be lowered up to \$0.47/kg syngas. However, this is much less environmentally friendly (0.83kg CO₂ eq./kg syngas) than the wind case.

...

In the energy mix case, it is important to reduce the “external H₂ additional cost” value (the cost of H₂ compression, storage and distribution when using external water electrolysis system) to change the supply of H₂ from the bicarbonate electrolyzer to external water electrolyzers. This is due to the expensive unit price of electricity and the higher cell voltage of the bicarbonate electrolyzer than that of water electrolyzers. If H₂ supply system is changed to water electrolysis, utility cost and electricity usage are decreased while material cost is increased. As a result, the “external H₂ additional cost” has high values of the Sobol indices for material cost, electricity usage, and CO₂ emission (See Fig. S20). In the case of the utility cost, OPEX, and break-even price, there is still a large amount of electricity for CO₂ conversion so that the “external H₂ additional cost” has a less impact. This is also confirmed by the violin plots in Fig. S24. While the solar and wind cases maintain low CO FE in most of samples, the energy mix case shows higher CO FE in some samples. High CO means most H₂ should be purchased rather than internal production by the bicarbonate electrolyzer. Except the “external H₂ additional cost”, the electrolyzer-related factors have high values of Sobol indices. Considering the bicarbonate electrolyzer is at its early development stage, there seems to have a large room for the improvement of the RSA process. Interestingly, CO₂ capture rate in the absorber is the major factor that determines OPEX. The higher capture rate, the more CO₂ is

converted and the more syngas is produced. This means higher operating cost is needed to produce more syngas. However, the ratio of those or syngas production price is highly sensitive to the FE improvement (the highest value of Sobol index for break-even price). Since water electrolysis is operated at much lower cell voltage than the bicarbonate electrolysis, water electrolysis can reduce the electricity use. This makes the “external H₂ additional cost” to have the highest value for the Sobol index of CO₂ emission.

The GSA results for the solar and wind cases have similar trend, except for CO₂ emission. Unlike the energy mix case that has high portion of fossil fuels for electricity generation, the solar and wind cases emit a very low amount of CO₂ for electricity generation. Between solar and wind, the CO₂ emission of solar energy is approximately 3.5 times higher than that of wind (41 vs. 12 g CO₂/kWh [5]). These make the different GSA result in environmental aspect. Therefore, the most sensitive factor for CO₂ emission is electricity usage reduction by changing H₂ supply for the energy mix case, electricity savings by improving the bicarbonate electrolyzer for the solar case, but reduction of uncaptured CO₂ released to air by increasing CO₂ capture rate in the absorber for the wind case, respectively.

Revision (Supporting Information): The purpose of modeling in this study is to find an economically and environmentally feasible CCU technology in order for the replacement of thermochemical CO₂ conversion. The three CCU processes considered in this study are thermal CO₂ conversion (RWGS, reverse water gas shift reaction), electrochemical CO₂ conversion (gas eCO₂R), and electrochemical bicarbonate conversion (RSA, reaction swing absorption) for syngas production. A few other technologies such as solid oxide electrolysis cell (SOEC) and dry methane reforming can be options for CO₂ utilizing syngas production. In particular, SOEC is one of the promising alternatives producing CO₂ oriented syngas because of its low cell potential and relatively high technological maturity [1]. Recently, large scale demonstrations for SOEC are in progress by Sunfire and Haldor-Topsoe. Although immediate problems such as high temperature durability must be solved for commercialization, SOEC needs to be assessed as a realistic solution for CO₂ conversion syngas production in following studies.

Flowsheets of these processes are illustrated in Fig. S13. Both the RWGS and gas eCO₂R processes require not only a stripping column for regeneration of amine but also pressure swing adsorption (PSA) system for separation of unreacted CO₂ and products (CO and H₂). Also, the reactor and PSA system are operated at different temperature in both the RWGS and gas eCO₂R processes so heat integration by heat exchanger network is needed. On the other hand, the RSA process uses neither the regeneration column nor PSA system, because unreacted CO₂ is still in the amine solution while low soluble products are in gaseous phase. Moreover, the RSA process has no need for heat integration, as it is operated at ambient temperature. The RWGS process produces CO only, so an on-site water electrolysis system (not in the flowsheet) is assumed to supply H₂ feed. Although the CO₂ and bicarbonate electrolyzers can produce H₂ as a by-product, the on-site water split system is also applied for the gas eCO₂R and RSA processes in order to satisfy H₂/CO ratio of 2 (desirable syngas for methanol) in the product

stream.

The process models are based on the literature as well as our experiment. The RWGS process is modeled, based on the result of the stable 80-hour operation of RWGS reaction from Sun et al. [2]. Cell voltage, current density and Faradaic efficiency (FE) of the gas eCO₂R are obtained from the experimental study of Liu, et al. [3] for over 3,000 hours. The PSA system for the RWGS and gas eCO₂R conversion processes is operated at 300°C [4] and its energy consumption, capital investment and operating cost are calculated by parameters from Jouny et al. [5]. When H₂ is produced by water electrolysis, a polymer electrolyte membrane (PEM) system is assumed to be constructed on the syngas production site [6]. The experiment data of bicarbonate electrolysis and chemisorption by trimethylamine (TREA) are used for the RSA process modeling. The relations among cell voltage, FE and current density in the bicarbonate electrolyzer are shown in Fig. S14.

...

LCAs with impact parameter values of materials, energies, etc. obtained from SimaPro V9.3 are conducted by Matlab, based on the process simulation results from Aspen Plus. The ReCiPe 2016(H) is used as a 'Cradle-to-Gate' method with the ecoinvent database V3.8 for the system boundaries in Fig. 4(c). For simplicity, transportation of raw materials (amines and water) to the production site is ignored for LCA. Because the three processes in this study have early technology maturity with different technology readiness levels, they are highly uncertain and lack of data availability [9–10]. Many LCA studies of CCU technology in the literature have been conducted for comparative assessment using 'Cradle-to-Gate' approach with functional unit of mass [11]. As a result, this study employed same methodology and conducted sensitivity analysis and scenario analysis.

As energy system has been changing from fossil to renewable and the bicarbonate electrolysis is currently at its early development, two scenarios are considered in this study: current and optimistic. In the optimistic scenario, i) the bicarbonate electrolyzer is improved, as seen in Fig. S14, ii) the water electrolysis system for on-site H₂ production is advanced [6], and iii) electricity generation cost is reduced [12]. Also, there are three cases in each scenario, based on energy sources for electricity generation: energy mix, solar, and wind. Thus, total 18 results (2 scenarios * 3cases * 3 processes) can be obtained.

Table S4 lists parameters values in the current and optimistic scenarios. The electrolyzer-related parameters are FE improvement and CD improvement for bicarbonate electrolysis and unit cost of bicarbonate electrolyzers. CD improvement means that the cell voltage at the same CD value will be reduced up to 1 V in the optimistic scenario (Fig. S14a). In the case of FE improvement, increasing up to 200 mA/cm² at the same FE value is assumed for the optimistic scenario (Fig. S14b). The unit cost of electrolyzer in the current scenario comes from the study of Jouny et al. [5].

...

The H₂ production by an on-site water electrolysis plant is based on the report of James et

al. [6]. Its cost consists of electricity cost and additional cost for fixed O&M cost, capital cost, etc. The total cost of them is considered a material cost for H₂ in the syngas production processes, because this study assumes H₂ purchase from the water electrolyzer as the internal H₂ procurement for the syngas production processes. The report [6] provides the amount of electricity consumption (54.3 vs. 50.2 kWh/kg H₂) and additional cost (\$1.74 vs. \$0.74/kg H₂) for current and optimistic scenarios.

...

The operating expenditure (OPEX) for the three processes are shown in Fig. S19. All three processes have similar OPEX in the current scenario, but the RSA outperforms in the optimistic scenario due to improved current density and CO FE at lowered cell voltage. In the current scenario, the optimal cell voltage of the bicarbonate electrolyzer is 3.1V for the energy mix case and 3.4V for solar and wind cases. The extremely high equipment cost of the bicarbonate electrolyzer is observed because of the low current density performance (42–72 mA/cm²). Higher cell voltage to increase current density can be applied, but this leads to low CO FE (41–51%) and high H₂ FE (49–59%). It means much of H₂ in the syngas comes from the bicarbonate electrolyzer at higher voltage than that in the on-site water electrolyzer. However, as the bicarbonate electrolysis technology will become mature in the future, approximately 3-fold higher current density with 90% CO FE at 0.3–0.4V lowered cell voltage is expected in the optimistic scenario, making substantial reduction of both CAPEX and OPEX. When the RSA process uses renewable energy only for electricity, syngas price in the optimistic scenario are \$0.56/kg syngas for solar and \$0.65/kg syngas for wind, respectively, that can economically compete with the fossil-fuel-based conventional technology.

...

As a result, the RSA process with the wind case in the optimistic scenario seems the best selection of syngas production from the economic and environment points of view: \$0.65/kg syngas with CO₂ emission of 0.27kg CO₂ eq./kg syngas. When solar energy is used, the break-even price can be lowered up to \$0.56/kg syngas. However, this is much less environmentally friendly (0.72kg CO₂ eq./kg syngas) than the wind case.

...

In the energy mix case, the improvement of CO FE, instead of current density, has the highest impact on CAPEX. Whereas increasing current density reduces CAPEX by the smaller bicarbonate electrolyzer area, the higher CO FE decreases CAPEX by the lower H₂ production rate in the bicarbonate electrolyzer. Therefore, CO FE improvement also affects the materials cost by purchasing more H₂ from the on-site water electrolysis system and the utility cost by reducing electricity use in the bicarbonate electrolyzer. In the case of CO₂ emission, the wind share increase is the most important factor as wind is the least CO₂ emitting energy source and has the higher variation of its share in energy mix (8.4–33.6%) than that of solar share (2.3–16.1%).

The solar and wind cases have the similar GSA results. The cost improvement of electricity generation by solar and/or wind plays a significant role in the economic and

environmental points of view. This is because the utility cost is considerably reduced from the current to the optimistic scenario (Fig. S19) and the renewable energies have a potential for further cost reduction in the future. An interest thing in the wind case is that the CO₂ capture rate in the absorber column has the highest Sobol index value to the CO₂ emission. GWP is much lower when wind energy is used (Fig. S20c) as wind is the cleanest energy source. Therefore, the importance of uncaptured CO₂ in the absorber that is released to the atmosphere is relatively high.

Original Sentence (Figures and Tables in Supporting Information):

Figure S14. Experiment data and future expectation of the bicarbonate electrolysis performance. (a) Current density curves of current and optimistic scenarios. (b) Faradaic efficiency curves of current and optimistic scenarios.

Figure S18. Capital investment of three processes under various energy sources. (a) Energy mix case. (b) Solar case. (c) Wind case.

Figure S19. Operating cost and break-even price of three processes under various energy sources. (a) Energy mix case. (b) Solar case. (c) Wind case.

Figure S20. LCA result of three processes under various energy sources. (a) RWGS process. (b) Gas eCO₂R process. (c) CO₂ RSA process.

Figure S21. GSA result of the RSA process in the energy mix case. (a) 1st order of Sobol index. (b) Total order of Sobol index.

Figure S22. GSA result of the RSA process in the solar case. (a) 1st order of Sobol index. (b) Total order of Sobol index.

Figure S23. GSA result of the RSA process in the wind case. (a) 1st order of Sobol index. (b) Total order of Sobol index.

Table S3. Parameters for techno-economic evaluation.

Parameter	Value
Plant Life (year)	20
Interest Rate (%)	7.0
Taxation (%)	38.9
Operating Time (day/year)	330
Labor Rate (\$/[hour·personnel])	40
Depreciation Method	Modified Accelerated Cost Recovery System for 10-year property

MEA Price (\$/kg)	3.0
TREA Price (\$/kg)	15.0
Process Water Price (\$/1000 gal)	0.8
O2 Price (\$/kg)	0.0
Electricity Price (\$/kWh)	Case Dependent
Natural Gas Fuel Price (\$/SCF)	5.0e-3
Steam Price (\$/kg)	0.014
Cooling Water Price (\$/kg)	2.9e-5
External H ₂ Production	
Electricity Consumption (kWh/kg)	54.3
Additional Cost (\$/kg)	4.18

Table S4. Parameters values used for global sensitivity analyses.

	Current	Optimistic
CO ₂ Capture Rate (%)	90	95
FE Improvement (V)	0	1
CD Improvement (V)	0	2
Unit Cost of Electrolyzer (\$/m ²)	919.7	300
Increase of Wind Share in Energy Mix (times)	1	4
Increase of Solar Share in Energy Mix (times)	1	7
Electricity Generation Improvement (%)	0	80
Improvement of Additional Cost of External H ₂ (%)	0	100

Revision (Figures and Tables in Supporting Information):

Figure S14. Experiment data and future expectation of the bicarbonate electrolysis performance. (a) Current density curves of current and optimistic scenarios. (b) Faradaic efficiency curves of current and optimistic scenarios.

Figure S18. Capital investment of three processes under various energy sources. (a) Energy mix case. (b) Solar case. (c) Wind case.

Figure S19. Operating cost and break-even price of three processes under various energy sources. (a) Energy mix case. (b) Solar case. (c) Wind case.

Figure S20. LCA result of three processes under various energy sources. (a) RWGS process. (b) Gas eCO₂R process. (c) CO₂ RSA process.

Figure S22. GSA result of the RSA process in the energy mix case. (a) 1st order of Sobol index. (b) Total order of Sobol index.

Figure S23. GSA result of the RSA process in the solar case. (a) 1st order of Sobol index. (b) Total order of Sobol index.

Figure S24. GSA result of the RSA process in the wind case. (a) 1st order of Sobol index. (b) Total order of Sobol index.

Table S3. Parameters for techno-economic evaluation.

Parameter	Value
Plant Life (year)	20
Interest Rate (%)	7.0
Taxation (%)	38.9
Operating Time (day/year)	330
Labor Rate (\$/[hour-personnel])	40
Depreciation Method	Modified Accelerated Cost Recovery System for 10-year property
MEA Price (\$/kg)	3.0

TREA Price (\$/kg)	15.0
Process Water Price (\$/1000 gal)	0.8
O2 Price (\$/kg)	0.0
Electricity Price (\$/kWh)	Case Dependent
Natural Gas Fuel Price (\$/SCF)	5.0e-3
Steam Price (\$/kg)	0.014
Cooling Water Price (\$/kg)	2.9e-5
On-site H ₂ Production by Water Electrolysis	
Electricity Consumption (kWh/kg)	54.3
Additional Cost (\$/kg)	1.74

Table S4. Parameters values used for global sensitivity analyses.

	Current	Optimistic
CO ₂ Capture Rate (%)	90	95
FE Improvement (mA/cm ²)	0	200
CD Improvement (V)	0	1
Unit Cost of Electrolyzer (\$/m ²)	919.7	300
Increase of Wind Share in Energy Mix (times)	1	4
Increase of Solar Share in Energy Mix (times)	1	7
Electricity Generation Improvement (%)	0	80
Improvement of Additional Cost of External H ₂ (%)	0	100

4. (Reviewer's Comment) *My original comment 13: I disagree with the authors that the SOEC based CCU technologies are not economically feasible and have a short-term CCU application. In fact, the Haldor-Topsoe process and the Sunfire co-electrolysis process are (near) commercial (they have much higher TRLs than the RWGS and the low temperature processes considered by the authors). Furthermore, I disagree with the authors that a large amount of heat required in the SOEC emits a lot of CO₂. These SOECs operate close to the thermoneutral voltage, thus the heat demand for the reaction itself is not so high. Pre-heating can be achieved by e.g., burning hydrogen instead of methane without additional CO₂ emissions. Proper heat integration can make SOECs very efficient for CCU applications now and in the future.*

(Authors' Response) Thank you for your detail comment and we apologize for the unclear statement in the original manuscript. We did not intend to say that SOEC based technologies are not economically feasible. We agree that SOEC based syngas production can be a promising option and aware Sunfire SOEC project (2.7 MW) and Haldor-Topsoe SOEC (500 MW). In the revised supporting information, we mentioned SOEC process as a potential option for realizing carbon neutral CCU technology.

Changes made:

Original Sentence: The purpose of modeling in this study is to find an economically and environmentally feasible CCU technology in order for the replacement of thermochemical CO₂ conversion. The three CCU processes considered in this study are thermal CO₂ conversion (RWGS, reverse water gas shift reaction), electrochemical CO₂ conversion (gas eCO₂R), and electrochemical bicarbonate conversion (RSA, reaction swing absorption) for syngas production. Solid oxide electrolysis cell (SOEC) can also be an option for syngas production [1], but SOEC requires a large amount of heat to be operated at high temperature (600–850°C) that causes a lot of CO₂ emission. Also, the major cell component for electrolyzer in SOEC system is an expensive ceramic material, such as yttria-stabilized zirconia, leading to large capital investment. Consequently, this study does not consider SOEC as an option.

Revision: The purpose of modeling in this study is to find an economically and environmentally feasible CCU technology in order for the replacement of thermochemical CO₂ conversion. The three CCU processes considered in this study are thermal CO₂ conversion (RWGS, reverse water gas shift reaction), electrochemical CO₂ conversion (gas eCO₂R), and electrochemical bicarbonate conversion (RSA, reaction swing absorption) for syngas production. A few other technologies such as solid oxide electrolysis cell (SOEC) and dry methane reforming can be options for CO₂ utilizing syngas production. In particular, SOEC is one of the promising alternatives producing CO₂ oriented syngas because of its low cell potential and relatively high technological maturity. Recently, large scale demonstrations for SOEC are in progress by Sunfire and Haldor-Topsoe. Although immediate problems such as high temperature durability must be solved for commercialization, SOEC needs to be assessed as a realistic solution for CO₂ conversion syngas production in following studies.

Responses to the Comments of the Reviewer 2

Review of the revised manuscript “Toward economical application of carbon capture and utilization technology with near zero carbon emission” of Lengie et al.

In the revised manuscript the authors did a dedicated exercise to answer the remarks of the reviewers. The literature has been updated and some of the assumptions that were not initially clear have been clarified. The quality of the manuscript has thus substantially increased. I think the manuscript is ready to be published after the authors cover a small concern I still have.

1. (Reviewer’s Comment) Concerning comment 3.

Although your TREA electrolyzer is at its early development stage that doesn’t exclude that you can add a brief discussion about the impact of the impurities on FEs. In fact, in the response letter you mention that NO_3^- shows large impact on FE. The changes made to the experimental section are not enough and the reader might reach the conclusion that these impurities are not being considered for upscaling steps. You could include the very same explanation and reference you provided to us to the main manuscript and, if you are considering to investigate these effects in future steps, mention in as future perspectives.

(Authors’ Response) Thank you for the comment. The authors have added a note on impurities with the reference in the conclusion.

Changes made:

Revision: It is worth to note that the impact of trace amounts of impurities, such as NO_3^- , on the bicarbonate electrolysis system should be investigated before upscaling the process.³⁹

Reviewer Comments, third round -

Reviewer #1 (Remarks to the Author):

I am satisfied with the technical corrections and additions to the manuscript. The paper can be accepted for publication.